# HIF-2α-pVHL complex reveals broad genotype-phenotype correlations in HIF-2α-driven disease

Daniel Tarade [1], Claire M. Robinson[1,2], Jeffrey E. Lee[1] & Michael Ohh [1,2]

It is definitively established that mutations in transcription factor HIF-2α are causative of both neuroendocrine tumors (class 1 disease) and polycythemia (class 2 disease). However, the molecular mechanism that underlies this emergent genotype–phenotype relationship has remained unclear. Here, we report the structure of HIF-2α peptide bound to pVHL-elongin B-elongin C (VBC) heterotrimeric complex, which shows topographical demarcation of class 1 and 2 mutations affecting residues predicted, and demonstrated via biophysical analyses, to differentially impact HIF-2α-pVHL interaction interface stability. Concordantly, biochemical experiments showed that class 1 mutations disrupt pVHL affinity to HIF-2α more adversely than class 2 mutations directly or indirectly via impeding PHD2-mediated hydroxylation. These findings suggest that neuroendocrine tumor pathogenesis requires a higher HIF-2α dose than polycythemia, which requires only a mild increase in HIF-2α activity. These biophysical data reveal a structural basis that underlies, and can be used to predict de novo, broad genotype-phenotype correlations in HIF-2α-driven disease.

[1] Department of Laboratory Medicine & Pathobiology, University of Toronto, 1 King's College Circle, Toronto, ON M5S 1A8, Canada. [2] Department of Biochemistry, University of Toronto, 661 University Avenue, Toronto, ON M5G 1M1, Canada. Correspondence and requests for materials should be addressed to M.O. (email: michael.ohh@utoronto.ca)

Recent discoveries have established that mutations in *EPAS1*, the gene encoding hypoxia-inducible factor (HIF)-2α transcription factor, are causative of familial and sporadic instances of polycythemia, as well as various neuroendocrine tumors[1–3]. *EPAS1*-linked polycythemia in the absence of neuroendocrine tumors is known as erythrocytosis type 4 (ECYT4; Mendelian Inheritance in Man (MIM) #: 611783). Patients who develop neuroendocrine tumors present with pheochromocytoma and paraganglioma (PPGL), which are closely related, norepinephrine-producing tumors arising from the neural crest[4]. Patients can present with either a single or multiple PPGLs. Presentation of multiple PPGLs is sometimes associated with concurrent somatostatinoma, a functional neuroendocrine tumor located in the duodenum, and polycythemia. This syndrome is known as Pacak-Zhuang syndrome[2,5]. In studies featuring large cohorts of patients, *EPAS1* mutations were identified in 5.7% of patients presenting with sporadic PPGLs (18/315, range of 2.3–12%), establishing *EPAS1* mutations as a major driver of PPGL[6–9].

*EPAS1* mutations have been shown, regardless of associated phenotype, to result in an increase in HIF-2α stability, by disrupting negative regulation via prolyl hydroxylase domain containing enzyme (PHD) and/or von Hippel-Lindau protein (pVHL) affinity to HIF-2α[2,10]. pVHL acts as the substrate-conferring component of an E3 ubiquitin ligase (pVHL/elongin BC/Cullin-2) and its interaction with HIF-2α is required for the rapid polyubiquitylation and proteasomal degradation of HIF-2α under normoxic conditions[11]. The interaction of pVHL with HIF-2α requires the hydroxylation of one of two conserved proline residues within the oxygen dependent degradation (ODD) domain of HIF-2α[11]. The hydroxylation of proline is catalyzed by PHDs and requires iron, ascorbate, α-ketoglutarate (αKG), and molecular oxygen as co-factors and co-substrates, thus allowing PHDs to function as oxygen-sensors[11]. Mutations in both *VHL* and *EGLN1* (encodes PHD2) are known to cause both polycythemia[12,13] and PPGL[14,15]. *VHL* mutation associated with PGL, polycythemia, and somatostatinoma has also been reported[16]. However, pheochromocytoma-associated *VHL* mutations have previously been suspected of triggering oncogenesis in a HIF-independent manner[17].

It has been noted that *EPAS1* mutations associated with ECYT4 are genetically distinct from those that are associated with neuroendocrine tumors[3]. However, to date, HIF-2α activating mutations associated with different phenotypes (ECYT4, Pacak-Zhuang Syndrome, sporadic PPGL) have not been studied in parallel. In addition, the studies of HIF-2α-driven disease have lacked structural insight into the interaction between HIF-2α and its negative regulator pVHL at the atomic level. Thus, the molecular basis for the emerging genotype–phenotype relationship in HIF-2α-driven disease has remained unclear.

## Results

**Class 1 and 2 HIF-2α diseases are driven by unique mutations**. As of 1 January 2018, 66 cases of HIF-2α-driven disease have been reported in the literature[2,5–10,18–35]. In one exceptional case, mutation of *EPAS1* was associated with central nervous system hemangioblastoma[36]. However, due to the rarity of HIF-2α-driven hemangioblastoma in reported literature, we focused on the role of HIF-2α in PPGL and polycythemia. Detailed clinical features of these reported cases are listed in Supplementary Data 1. We have identified four classes of disease (Table 1); patients who present with PPGL in conjunction with somatostatinoma and polycythemia (class 1a), patients who present with PPGL and polycythemia (class 1b), patients who present with PPGL alone (class 1c), and patients who present with polycythemia alone (class 2). Others have also categorized diseases associated with *EPAS1* mutations in a similar fashion[3], and class 2 disease has been previously described as ECYT4 while classes 1a and 1b have been jointly described as Pacak–Zhuang syndrome.

All HIF-2α mutations described to date are heterozygous and the majority are single missense mutations. There are a few reported cases of patients with multiple missense mutations, with both mutations affecting the same allele, or small in-frame deletions (Table 1). With one exception (patient 15, Supplementary Data 1), all class 1 mutations are defined as being somatic. Conversely, class 2 disease patients present with germline mutations and as families in 60% of reported cases (Table 1).

Regardless of the associated clinical phenotype, over 90% of reported HIF-2α mutations are located between amino acid residues 519 and 544 (Fig. 1a). Within this region, there is a clear topographical distribution of mutations associated with different phenotypes. Mutations associated with class 1 disease are typically located between amino acid residues 529 and 532, which contain proline 531, the primary hydroxylation site of HIF-2α (Fig. 1b). Conversely, class 2 mutations are found exclusively

### Table 1 Clinical features of HIF-2α-driven disease

| | Class 1a (*n* = 6) | Class 1b (*n* = 12) | Class 1c (*n* = 20) | Class 2 (*n* = 28) |
|---|---|---|---|---|
| *Disease penetrance* | | | | |
| PPGL (%) | 100 | 100 | 100 | 0 |
| SOM (%) | 100 | 0 | 0 | 0 |
| Polycythemia (%) | 100 | 100 | 0 | 100 |
| *Age of disease onset (years)* | | | | |
| PPGL | 28.00 ± 13.2 | 22.42 ± 16.0 | 49.26 ± 16.7 | – |
| SOM | 34.83 ± 13.4 | – | – | – |
| Polycythemia | 5.00 ± 6.1 | 1.55 ± 2.2 | – | 28.21 ± 14.3 |
| *Heritability* | | | | |
| Family history (%) | 0 | 8.33 (1 familial case, 11 sporadic cases) | 0 | 60 (9 familial cases, 6 sporadic cases) |
| *Type of mutation* | | | | |
| Heterozygous (%) | 100 | 100 | 100 | 100 |
| Single missense (%) | 100 | 91.67 | 85 | 100 |
| Multiple missense (%) | 0 | 8.33 | 5 | 0 |
| Microdeletion (%) | 0 | 0 | 10 | 0 |

Includes information on all reported cases before 1 January 2018. Average age of diagnosis only includes cases where an exact age is given. Values are represented as mean ± SD
PPGL: pheochromocytoma and paraganglioma, SOM: somatostatinoma

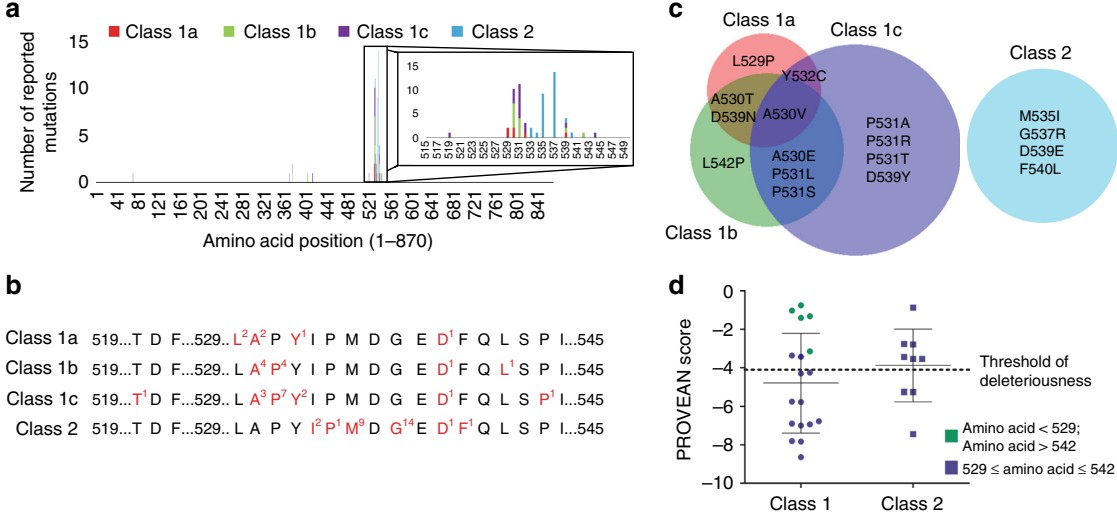

Fig. 1 Class 1 and class 2 HIF-2α-driven diseases are associated with different mutations. **a** Frequency of HIF-2α missense mutations across the primary amino acid sequence. An inset of amino acids 515-550 is provided. **b** The majority of HIF-2α mutations are missense mutations located between amino acids 519 and 545. Residues that have been reported to be mutated are highlighted in red. The superscript indicates how often the residue has been mutated. **c** A Euler Diagram highlighting the proportion of unique mutations associated with each disease class. Examples of mutations associated with each disease class are listed. **d** A PROVEAN score was determined for each missense EPAS1 mutation. The more negative a score, the more damaging a mutation is predicted to be. A mutation with a score lower the −4.1 is predicted to be deleterious with high specificity. Error bars indicate SD

between residues 533 and 540 (Fig. 1b). However, this topographical segregation is not absolute, as mutations of D539 are found in patients of all disease classes. Furthermore, class 1 mutations are also found outside of the 529-532 hotspot, including nearby mutations of T519, L542, and P544 and more distant mutations of amino acids S71, M368, and F374 (Fig. 1a). Interestingly, there is no overlap in the specific mutations associated with class 1 and class 2 disease, as the mutations affecting D539 are not the same among class 1 and class 2 (D539N in class 1a and 1b, D539Y in class 1c, D539E in class 2; Fig. 1c). However, there is substantial overlap in the mutations that give rise to the various subclasses of class 1 disease. For example, the A530V mutation has been reported in patients with class 1a, class 1b, and class 1c disease (Fig. 1c). Thus, the broad phenotypic differences between class 1 and class 2 disease stem from genetic differences while the differences between subclasses of class 1 disease might be related to the presence of additional mutations or the occurrence of *EPAS1* mutations during embryonic development or later in life.

Our first approach for testing the differences between class 1 and class 2 *EPAS1* mutations was to employ a suite of mutation prediction software, including PolyPhen-2, SIFT, MutationTaster2, and PROVEAN. Of the tested mutation prediction software, not one was able to consistently distinguish between class 1 and class 2 mutations. MutationTaster2 annotated all class 1 mutations as disease causing while SIFT and Polyphen-2 classified 77.8% (14/18) and 83.3% (15/18) of class 1 mutations as probably damaging, respectively. The three aforementioned mutation prediction software classified 100% of class 2 mutations as probably damaging (Supplementary Table 1). However, PROVEAN annotated 61.1% of class 1 mutants as deleterious (mean PROVEAN score of −4.80) and only 33.3% of class 2 mutants as deleterious (mean PROVEAN score of −3.43) when a threshold of −4.1 was used (Fig. 1d). Interestingly, the five class 1 mutations with the least negative PROVEAN score are those found either C-terminal or N-terminal of amino acids 529–542 (Fig. 1d). Due to those five class 1 mutations, there is a greater percentage of 'benign' mutations present within class 1 disease than present within class 2 disease. Thus, mutation prediction

software was ultimately inadequate at predicting disease class, as mutations from both class 1 and class 2 were found with similar PROVEAN scores.

**HIF-1α and HIF-2α peptides bind to pVHL via a similar motif.** The failure to distinguish class 1 mutations from class 2 mutations via in silico approaches provided the impetus for us to pursue a biochemical approach, guided by structural insight obtained via X-ray crystallography. As HIF-2α has not been co-crystallized with pVHL, it is not formally known whether it shares the same binding motif as HIF-1α, thereby potentially limiting the utility of existing HIF-1α-pVHL structures[37,38]. There are indeed several amino acids in the immediate vicinity of the primary hydroxylation site that are not conserved between HIF-1α and HIF-2α but are highly conserved evolutionarily (Fig. 2a). The two notable sites of non-conservation are HIF-1α Met561/HIF-2α Thr528 and the insertion of HIF-2α Gly537, which is not present in HIF-1α. Intriguingly, G537 is the most commonly mutated residue in HIF-2α driven disease while the substitution of the non-polar, sulfur-containing Met residue with a polar Thr residue supports the notion that HIF-2α peptide might not bind pVHL with an identical motif as HIF-1α. Further, attempts at modeling the HIF-2α-pVHL interaction and predicting the effect of mutations on pVHL affinity to HIF-2α via computational approaches failed to differentiate class 1 and 2 mutations[39].

We purified pVHL-elongin B-elongin C (VBC) trimeric complex to homogeneity from bacterial sources (Supplementary Figure 1). We pursued an unbiased sparse matrix crystallography approach rather than attempt co-crystallization with reagents used to co-crystallize HIF-1α with VBC since we were unsure whether HIF-2α would bind to pVHL with a motif similar to that of HIF-1α. A complex of hydroxylated HIF-2α peptide (523–541) bound to VBC trimer was co-crystallized and the structure was determined at 2.0Å resolution (PDB code: 6BVB). Data collection and structure determination statistics are presented in Supplementary Table 2. Clear electron density was observed for residues 527–540 of the hydroxylated HIF-2α peptide (Supplementary Figure 2). The HIF-2α peptide binds to pVHL as an extended strand with N-terminal (residues 527–534) and C-terminal

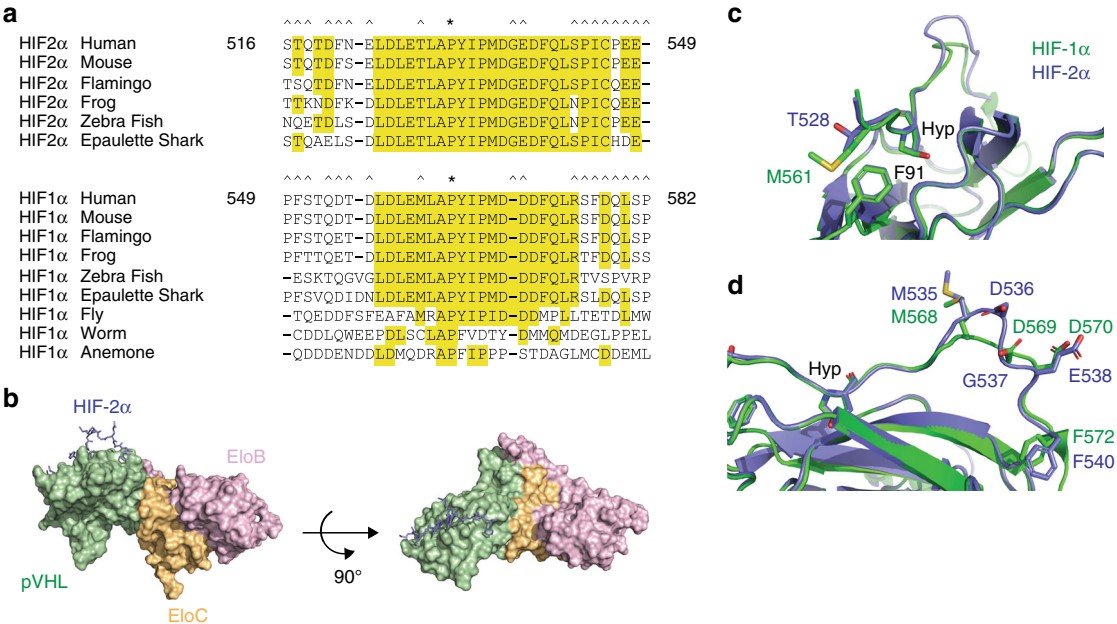

**Fig. 2** HIF-1α and HIF-2α peptides bind VBC complex with a similar motif. **a** Alignment of HIF-1α and HIF-2α proteins from various species. *denotes the primary hydroxylation site. ^denotes residues that are not conserved between human HIF-1α and HIF-2α. Residues shaded in yellow are conserved in at least two out of three species for that protein. **b** Model of pVHL-elongin B-elongin C-HIF-2αOH complex. HIF-2α is shown as a stick model in light blue. pVHL (green)-elongin B (pink)-elongin C (yellow) are displayed as a van der Waal surface. **c, d** Crystal structures of pVHL-elongin B-elongin C-HIF-1αOH complex (1LQB) and pVHL-elongin B-elongin C-HIF-2αOH complex are superimposed. Carbon is shown in green (HIF-1α) or light blue (HIF-2α), oxygen is shown in red, nitrogen is shown in blue, and sulfur is shown in yellow

(residues 539–540) pVHL contact sites (Fig. 2b). The root-mean square displacement (RMSD) of Cα atoms between a previously reported *apo* VBC structure (PDB code: 1VCB)[40] and VBC in our co-crystal structure is 0.94 Å, suggesting that VBC does not undergo a conformational change upon binding of HIF-2α peptide. A lack of conformational change was also noted upon HIF-1α peptide binding[37]. Further, RMSD of Cα atoms of the HIF-2α peptide compared with HIF-1α peptide from previously reported co-crystal structures also revealed rather minor changes in conformation (0.78 Å with 1LM8, 0.85 Å with 1LQB). The substitution of the amino acid three residues N-terminal of the primary hydroxylation site (Met561/Thr528) had no effect on the local conformation, with both residues packed against pVHL F91 (Fig. 2c). The insertion of Gly537, seen in HIF-2α, results in an increased sinusoidal nature of the HIF-2α peptide but that only results in local conformational changes (Fig. 2d).

**Class 1 mutations disrupt significant pVHL-HIF-2α contacts.** The majority of contacts between HIF-2α and pVHL occur at the N-terminal pVHL contact site through hydrogen bond and van der Waals interactions. The hydroxylated P531 (Hyp531) is almost completely buried in a highly complementary pVHL pocket composed of residues W88, Y98, I109, S111, Y112, H115, and W117 (Fig. 3a and Supplementary Figure 3). The HIF-2α Hyp531 hydroxyl is coordinated by two H-bonds to the sidechain of pVHL S111 and H115, and a third H-bond is formed between the Hyp531 carbonyl oxygen atom and the sidechain of pVHL Y98 (Fig. 3a). The carbonyl oxygen of HIF-2α E527 hydrogen bonds to the side chain of both pVHL N67 and pVHL R69 (Fig. 3a). Furthermore, two hydrogen bonds are formed between the mainchain amide nitrogen and carbonyl oxygen atoms of HIF-2α Y532 and pVHL H110 (Fig. 3a). Residues 535–538 form a kink that does not interact with pVHL (Fig. 3a and Supplementary Figure 3). F540 serves as a C-terminal anchor, contacting

pVHL G106, that re-establishes contact between pVHL and HIF-2α (Fig. 3a).

Interestingly, the majority of reported class 2 mutations (82%) are localized to the kink region whereas class 1 mutations affect residues making contact with pVHL (Fig. 3b). Mutations of M535 and G537 are the most common class 2 mutations (Fig. 1b) and are localized to this kink region (Fig. 3a, b). Temperature factor analysis reveals that residues C-terminal of the kink region are mobile, and importantly, that the residues mutated in class 1 disease have a lower temperature factor and bind in a more rigid manner (Fig. 3c). Thus, a prediction based on our structural analysis is that class 1 mutations would have a greater negative impact on pVHL-HIF-2α interaction than class 2 mutations.

**Class 1 mutations strongly effect pVHL affinity.** To test our hypothesis that class 1 mutations have a more adverse effect on pVHL affinity (referring specifically to the strength of interaction between HIF-2α peptide or protein and pVHL), we conducted both steady-state and kinetic binding experiments. Previous ELISA-based experiments with polycythemia-associated HIF-2α mutants suggested that class 2 mutants had no effect on VBC affinity[31]. Similarly, we show that steady-state ELISA experiments conducted with HIF-2αOH (amino acids 523–541) peptides (i.e., hydroxyl group synthetically added to P531) and purified VBC complex suggested that both class 1 and class 2 mutations had negligible or minimal effect on pVHL affinity (Fig. 4a), with the exception of the P531A mutant, which completely abolished binding to VBC as expected. However, we found that class 1 mutations, in general, have a more deleterious effect on pVHL affinity than class 2 mutations when analyzed via pull-down experiments using in vitro transcribed and translated (IVTT) pVHL under both stringent and mild buffer conditions (Fig. 4b and Supplementary Figure 4). Intriguingly, HIF-2αOH A530V class 1 peptide consistently bound to pVHL or VBC with an

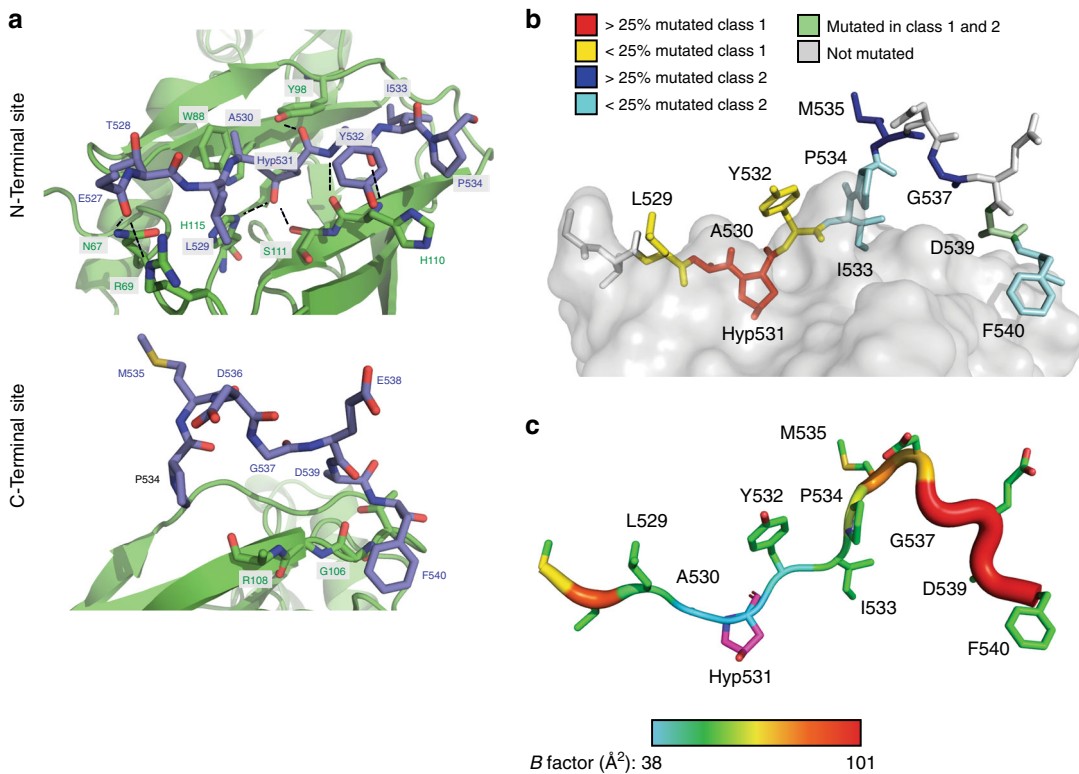

**Fig. 3** Class 1 mutations impact residues important for pVHL affinity to HIF-2α. **a** HIF-2α is shown as a stick model and pVHL is modeled as a cartoon with key residues displayed as stick models. Carbon is shown in green (pVHL) or light blue (HIF-2α), oxygen is shown in red, nitrogen is shown in blue, and sulfur is shown in yellow. Hydrogen bonds are denoted by dashed lines. (top panel) Interaction of HIF-2α residues 527-534 with pVHL. (bottom panel) Interaction of HIF-2α residues 534–540 with pVHL. Backbone atoms for pVHL residues G106 and R108 are labeled. **b** The HIF-2α peptide is shown as a stick model and the pVHL complex is displayed as a van der Waal surface (gray) set to 50% transparency. Mutations reported in HIF-2α driven disease are color-coordinated according to mutation frequency. Residues not reported to be mutated are shown in gray. **c** The residues of the HIF-2α peptide are colored according to temperature factor ($B$ factor). As the $B$ factor increases, the color transitions from blue to red and the peptide backbone (modeled as a putty) increases in diameter. A high $B$ factor is indicative of increased flexibility and mobility. $B$ factor values range from a low of 38.13 (Hyp531) and to a high of 100.28 (F540)

affinity similar to that of HIF-2αOH WT peptide and other class 2 mutants, such as M535I.

To gain further quantitative insight, we next analyzed the effect of HIF-2α mutations on the kinetics of HIF-2α peptide binding to pVHL via biolayer interferometry (BLI)[41], which measures the rate of association and dissociation of an analyte molecule (purified VBC complex) to and from an immobilized bait molecule (biotinylated HIF-2αOH peptide). Consistent with our steady-state pull-down experiments, BLI experimentation revealed that HIF-2αOH WT peptide had the highest affinity for VBC ($K_d = 196$ nM; $k_a = 4.1 \times 10^4$ Ms$^{-1}$; $k_d = 7.8 \times 10^{-3}$ s$^{-1}$; Fig. 4c, Table 2, and Supplementary Figure 5). No binding between HIF-2α P531A peptide and VBC was detected, which validated the stringency of our experimental conditions. Most mutant HIF-2αOH peptides displayed decreased binding to VBC, consistent with steady-state pull down experiments (Table 2). Moreover, class I mutants increased the rate of dissociation, as highlighted in a rate plane with isoaffinity diagonals (RaPID) plot (Fig. 4d). The four mutations with the most deleterious effect on pVHL affinity all belonged to class 1 (L529P, A530T, P531A, Y532C; Fig. 4d, Table 2). The L529P mutation possibly results in steric clash with pVHL N67, resulting in a disruption of its hydrogen bond to HIF-2α E527. Y532 normally packs against the sidechain of pVHL H110 and the Y532C mutation would result in diminished van der Waals interaction. The P531A mutation would result in the loss of important hydrogen bonds, shown in Fig. 3a. Interestingly, and consistent with our

steady-state observations, the A530V ($K_d = 219$ nM) and A530T ($K_d = 513$ nM) mutations had different effects on pVHL affinity, despite affecting the same residue. The A530V HIF-2αOH peptide dissociated more slowly from VBC than WT ($k_d = 5.8 \times 10^{-3}$ s$^{-1}$) whereas the A530T HIF-2αOH peptide ($k_d = 18 \times 10^{-3}$ s$^{-1}$) behaved similarly to the other class 1 mutant peptides. Due to the extensive hydrogen bond network involving HIF-2α Hyp531, it is possible that the polar nature of the A530T mutation results in a degree of interference. Specifically, A530 points toward pVHL Y98, which hydrogen bonds to the Hyp531 carbonyl oxygen atom (Fig. 3a). Hydrogen bonding between the A530T mutant residue and pVHL Y98 may disrupt the Hyp531 pocket, promoting dissociation of pVHL. The decreased dissociation rate seen with the A530V (Table 2) mutation lends a degree of support to this argument, as the nonpolar mutation would possibly result in increased van der Waals interactions with the adjacent pVHL W88 residue without possibly hydrogen bonding to pVHL Y98.

Interestingly, the G537R class 2 mutation ($K_d = 348$ nM; $k_a = 3.0 \times 10^4$ Ms$^{-1}$; $k_d = 10 \times 10^{-3}$ s$^{-1}$) had a near identical effect on the kinetics of binding to VBC as did the F540L class 2 mutation ($K_d = 345$ nM; $k_a = 3.1 \times 10^4$ Ms$^{-1}$; $k_d = 10 \times 10^{-3}$ s$^{-1}$), despite being localized to the kink region that does not make any significant contacts with pVHL. However, based on the structure of HIF-2α peptide bound to pVHL (Fig. 3a), mutation of G537 to a bulkier amino acid (tryptophan or arginine; class 2 mutations) is predicted to result in steric clash with pVHL R108. Such a

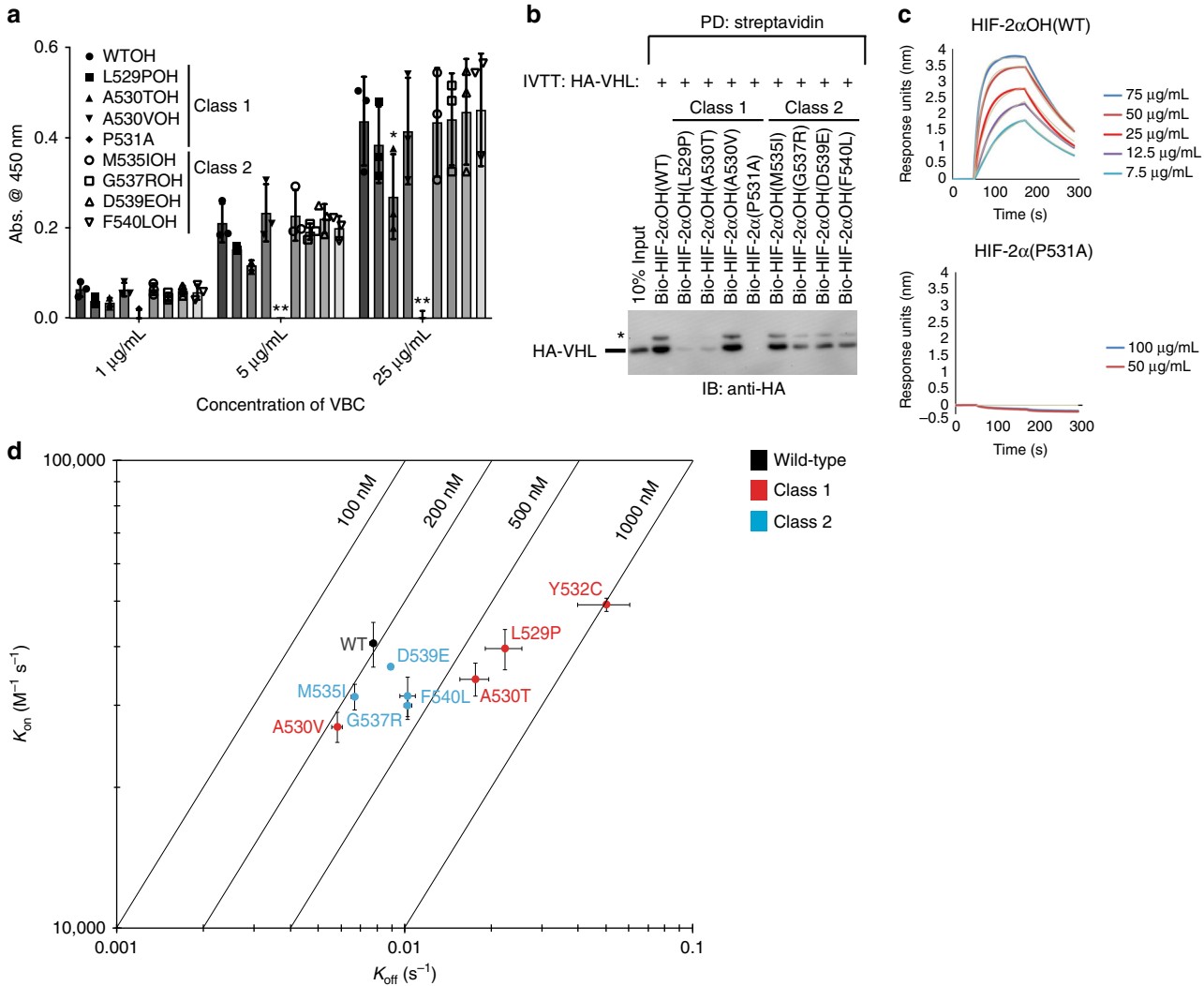

**Fig. 4** Affinity of HIF-2α mutants for pVHL is correlated with disease phenotype. **a** Biotinylated HIF-2α peptides (523–541) were immobilized on a streptavidin-coated 96-well plate. Binding to various concentrations of pVHL-elongin B-elongin C (VBC) complex was assayed via ELISA with an anti-eloB antibody. Absorbance was read at 450 nm. Values represent mean of three independent experiments performed in duplicate ± SEM. *$p < 0.05$; **$p < 0.01$. **b** Biotinylated HIF-2αOH peptides (523–541) were immobilized on streptavidin- agarose beads and incubated with in vitro transcribed and translated (IVTT) pVHL. Streptavidin beads were pulled down (PD) and levels of HA-tagged pVHL were visualized via immmunoblotting (IB). **c**, **d** Biolayer interferometry kinetic analysis of VBC complex binding to biotinylated HIF-2α peptides (523–541). **c** Biotinylated peptides were coupled to streptavidin-coated biosensors and monitored for binding to VBC complex at the indicated concentrations. The data were analyzed based on a 1:1 binding model using the BLItz Pro software with the fitted curves shown as gray lines. Sensorgrams are representative of three experiments conducted with independently purified protein. **d** Rate plane with Isoaffinity Diagonals (RaPID) plot highlighting the kinetic parameters of VBC complex binding to HIF-2α peptides. Values represent mean of three experiments conducted with independently purified protein ± SEM. * denotes uncharacterized band

mutation would disrupt the kink region and result in displacement of the C-terminal F540 residue and loss of stabilizing contacts. This is an important observation as G537 is not conserved in HIF-1α and is also the most commonly mutated residue in HIF-2α driven disease (Fig. 1a, b). Other class 2 mutant peptides (M535I, D539E) bound to VBC with little defect, as compared to WT (Fig. 4d, Table 2, and Supplementary Figure 5). Thus, although BLI kinetic analysis revealed a distinguishable kinetic trend between class 1 and class 2 mutations, the disease class associated with HIF-2α mutations could not be absolutely predicted solely by determining affinity for pVHL due to outliers like A530V.

**A subset of class 1 HIF-2α mutations abrogate hydroxylation**. Considering that prolyl hydroxylation is indispensable for HIF-2α binding to pVHL, we next examined the notion that the A530V

mutation negatively influences hydroxylation of HIF-2α via PHD2, which would attenuate HIF-2α A530V binding affinity for pVHL. We conducted an in vitro hydroxylation assay utilizing unmodified peptides (amino acids 523–542) incubated with purified $HIS_6$-PHD2 (catalytic core, amino acids 181–426). As expected, HIF-2α WT peptide bound robustly to pVHL following $HIS_6$-PHD2 treatment (Fig. 5a). Similar to using synthetically hydroxylated peptides (see Fig. 4b), M535I class 2 mutant still retained affinity for pVHL following incubation with $HIS_6$-PHD2 (Fig. 5a). However, A530V peptide was only minimally able to bind pVHL, much like the A530T mutant (Fig. 5a). These results suggest that the A530V mutation impedes PHD2-mediated hydroxylation of P531, which negatively impacts HIF-2α binding affinity for pVHL.

Due to the importance of L574 for hydroxylation of HIF-1α[42], the fact that this residue is conserved in HIF-2α (L542), and the

observation that the residue is mutated in class 1 disease (L542P, patient 7 in Supplementary Data 1), we posited that this mutation specifically affects hydroxylation but not pVHL affinity per se, similar to the A530V class 1 mutation. As predicted, HIF-2α L542P peptide failed to bind pVHL following incubation with $HIS_6$-PHD2, which was similarly observed with HIF-2α WT peptide lacking L542 (523–541; Fig. 5b). However, synthetic addition of a hydroxyl group to P531 in HIF-2αOH L542 peptide rescued binding to pVHL with an affinity similar to HIF-2αOH WT peptide, confirming that the L542P mutation specifically abolishes PHD2-catalyzed hydroxylation (Fig. 5b). These results demonstrate that a subset of class 1 mutations specifically disrupt PHD2-mediated regulation, which ultimately leads to a severe reduction in pVHL binding. In addition, these results prove that

the L542 residue is absolutely required for hydroxylation, despite published methodology suggesting otherwise[2].

The presence of class 1 HIF-2α mutants that bind pVHL like WT when synthetically hydroxylated but fail to be hydroxylated by PHD2 suggested to us that class 1 and class 2 mutations may differentially affect PHD2 binding. To first test this hypothesis, we performed steady-state pull-down experiments with HIF-2α peptides (523–542) and purified $HIS_6$-PHD2 (181–426). To stabilize the interaction between enzyme and substrate, it was necessary to include $MnCl_2$ and N-oxalylglycine (NOG) in the binding buffer. $Mn^{2+}$ and NOG substitute for $Fe^{2+}$ and αKG, respectively, which stabilizes the HIF-2α-PHD2 interaction while preventing enzymatic turnover. A similar strategy was employed during co-crystallization of HIF-1α peptide with PHD2[43]. We observed that all tested HIF-2α mutations result in decreased affinity for PHD2 (Supplementary Figure 6a). However, the M535I class 2 mutant was not as deleterious in this regard, which is consistent with its ability to be hydroxylated by PHD2 in vitro (Fig. 5a). Next, we co-transfected HEK293a cells with 3xFLAG-HIF-2α ODD and HA-PHD2 constructs. Immunoprecipitation of 3xFLAG revealed that all HIF-2α mutant ODD constructs bound less PHD2 than WT HIF-2α ODD (Supplementary Figure 6b). The increased binding of the M535I mutant to PHD2 was not observed in this experiment. One important caveat is that the HIF-2α ODD constructs contain both the P405 and P531 sites. Thus, the residual PHD2 binding observed among HIF-2α mutants may partially be driven by the P405 site, which may mask differential PHD2 binding among the HIF-2α mutant ODD constructs.

To further evaluate the negative effect of HIF-2α mutations on pVHL affinity, we transiently expressed full-length HIF-2α (WT and representative class 1 and class 2 mutants) in combination with FLAG-pVHL in human embryonic kidney epithelial HEK293a cells that express endogenous PHD2. We observed that under this experimental condition, class 1 mutants bound lower levels of pVHL than class 2 mutants (Fig. 5c). Notably, A530V class 1 mutant bound poorly to pVHL in our cellular system, which corroborated our earlier in vitro binding assay following in vitro PHD2-mediated hydroxylation (Fig. 5a). These results collectively support the notion that the observed

**Table 2 Kinetic parameters of HIF-2α peptides binding to purified VBC complex**

| Peptide | VBC complex | | |
|---|---|---|---|
| | $K_d$ (nM) | $k_a$ (x$10^4$) (Ms$^{-1}$) | $k_d$ (x$10^{-3}$) (s$^{-1}$) |
| HIF-2αOH (wild-type) | 196 ± 17 | 4.1 ± 0.4 | 7.8 ± 0.2 |
| HIF-2α (wild-type) | No binding | | |
| HIF-2αOH (L529P) | 558 ± 31 | 4.0 ± 0.4 | 22 ± 3 |
| HIF-2αOH (A530T) | 513 ± 21 | 3.4 ± 0.3 | 18 ± 2 |
| HIF-2αOH (A530V) | 219 ± 15 | 2.7 ± 0.2 | 5.8 ± 0.2 |
| HIF-2α (P531A) | No binding | | |
| HIF-2αOH (Y532C) | 998 ± 200 | 4.9 ± 0.2 | 50 ± 10 |
| HIF-2αOH (M535I) | 216 ± 9 | 3.1 ± 0.2 | 6.7 ± 0.2 |
| HIF-2αOH (G537R) | 348 ± 13 | 3.0 ± 0.2 | 10 ± 0.3 |
| HIF-2αOH (D539E) | 253 ± 12 | 3.6 ± 0.05 | 9.0 ± 0.1 |
| HIF-2αOH (F540L) | 345 ± 29 | 3.1 ± 0.3 | 10 ± 0.6 |

Values represent mean of three experiments conducted with independently purified protein ± SEM
$K_d$: affinity constant, $k_a$: association rate, $k_d$: dissociation rate

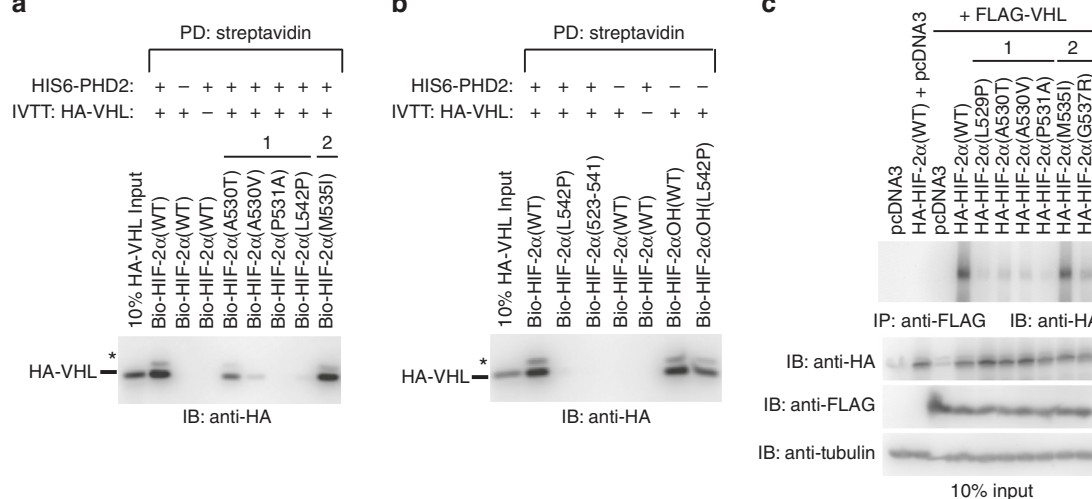

**Fig. 5** A subset of class 1 mutations specifically disrupts PHD2-mediated hydroxylation. **a**, **b** Biotinylated HIF-2α peptides (residues 523–542) were immobilized on streptavidin- agarose beads and incubated with $HIS_6$-PHD2 for 1 h at room temperature prior to incubation with in vitro transcribed and translated (IVTT) pVHL. Streptavidin beads were pulled down (PD) and levels of HA-tagged pVHL were visualized via immunoblotting (IB). **c** HEK293a cells were co-transfected with FLAG-VHL and HA-HIF-2α constructs. FLAG was immunoprecipitated (IP) and levels of HA-tagged HIF-2α were visualized via immunoblotting (IB). * denotes uncharacterized band

differential impact on hydroxylation via PHD2 and/or binding affinity to pVHL by class 1 versus class 2 HIF-2α mutations underlies the emerging genotype-phenotype relationships seen in patients.

## Discussion

Loss-of-function mutations in *VHL*, which encodes the tumor suppressor pVHL, cause VHL disease, an autosomal dominant cancer disorder with predisposition to central nervous system hemangioblastoma, PPGL, and clear cell renal cell carcinoma (ccRCC) as well as polycythemia, and the majority of sporadic cases of hemangioblastoma and ccRCC[44]. These diseases are thought to be driven, in part or wholly, by deregulation of HIF-α, the best characterized target of pVHL-mediated negative regulation[17]. Concordantly, it is now known that mutations in *EPAS1*, which encodes HIF-2α, also cause polycythemia, PPGL, and somatostatinoma. Upon analyzing every reported case (66 as of 1 January 2018) of *EPAS1* mutations in the aforementioned diseases, we propose the following classification system for HIF-2α-driven disease with class 1 disease featuring PPGL, which is subcategorized into class 1a featuring PPGL with somatostatinoma and polycythemia, class 1b featuring PPGL and polycythemia (classes 1a and 1b are also jointly known as Pacak-Zhuang syndrome), and class 1c featuring only PPGL. Class 2 disease features exclusively polycythemia, which has been previously described as ECYT4 (Table 1). A recent publication examined a cohort of patients with congenital cyanotic heart disease and PPGL. Of the five individuals, four were found to have somatic *EPAS1* mutations, all of which resulted in mutation of either amino acid residue A530 or P531 (P531S (x2), P531R, A530P)[45]. Thus, continued study of patients with PPGL affirms the emergence of *EPAS1* mutations as a major driver of neuroendocrine tumor pathogenesis and solidifies the importance of systemic hypoxia or pseudohypoxia in PPGL tumorigenesis.

Co-crystal structure of hydroxylated HIF-2α peptide encompassing residues 523–541, which cover over 90% of bona fide disease-causing *EPAS1* mutations identified to date, bound to VBC complex revealed that class 1 mutations affect residues contacting pVHL while the vast majority of reported class 2 mutations are localized to a non-contacting kink region (residues 535–538; Fig. 3b). This includes mutations of G537, which account for 50% of class 2 mutations. Further, residues C-terminal of the kink region, despite making contact with pVHL (Supplementary Figure 3), are more mobile than residues proximal to Hyp531, as indicated by temperature factor analysis (Fig. 3c). When considering both the kink region and the mobile C-terminal region, almost 90% of class 2 mutations affect residues predicted to contribute marginally to pVHL-HIF-2α interface stability. Conversely, over 80% of class 1 mutations affect residues that make substantial contact with pVHL (i.e. residues 529–532). In keeping with these structural data-based predictions, we found that the HIF-2α peptides with the lowest affinity for pVHL are indeed class 1 mutants (L529P, A530T, P531A, Y532C; Fig. 4).

Interestingly, the trend of greater loss of pVHL affinity associated with class 1 mutants was apparent when conducting kinetic binding experiments with purified VBC (Fig. 4d) but not easily appreciated when conducting steady-state ELISA experiments with VBC (Fig. 4a). In light of the BLI-measured dissociation constants ($K_d$) for WT and mutant HIF-2αOH peptides (200–500 nM; Table 2), it is clear that the minimum concentrations required to obtain a detectable signal during an ELISA experiment (575 nM) are saturating. As previously mentioned, prior attempts to characterize class 2 mutants by ELISA failed to identify any change in VBC affinity, when compared to WT HIF-2αOH peptide[31]. These observations highlight the importance of

kinetic binding analysis for studying disease-associated HIF-2α mutations, which has not been previously reported.

Although the majority of class 2 mutations are localized to the non-contacting kink region or flexible C-terminal region, mutations of I533 and P534 are also reported to cause polycythemia in the absence of neuroendocrine tumor development (Fig. 1b, Supplementary Data 1). These residues are proximal to Hyp531, make contacts with pVHL, and are predicted to contribute to the stability of the pVHL-HIF-2α interface. In particular, I533 binds pVHL more deeply than any other residue, aside from Hyp531 (Supplementary Figure 3, 7). However, the disease-causing mutations affecting these residues, I533V and P534L, are conserved mutations that can be accommodated at the pVHL-HIF-2α interface without introducing steric clash and with the loss of only minimal van der Waal interactions, which likely explains why these mutations are associated with class 2 disease phenotype despite affecting residues that make significant contact with pVHL. As a corollary, a prediction is that a bulky substitution at either of these sites, such as an I533F (C.1597 A > T) or P534R (C.1601C > G) missense mutation, would introduce steric clash that would destabilize the HIF-2α-pVHL binding interface. Although not yet reported in literature, any patient confirmed to carry these de novo mutations should be monitored for neuroendocrine tumors, as they would be predicted to have or develop class 1 disease.

A limitation of our study is the inability to provide a mechanistic understanding for the sub-classes of HIF-2α-driven disease, namely class 1a, 1b, and 1c disease. Whereas no mutation has been shown to cause both class 1 and class 2 disease, several mutations have been shown to cause two or more of the sub-classes of class 1 disease (Fig. 1c). Thus, a genotype–phenotype study of this phenomenon was not possible. However, it is possible to speculate on a possible mechanism that underlies the presence of sub-classes of class 1 disease based on the timing of mutations during embryonic development, which would have implications on the cell lineages affected. It has been previously shown that HIF-2α activity is inhibited in embryonic stem cells via a titratable repressor[46]. Increased expression of HIF-2α is thought to be required to overcome this form of negative regulation[46], which has been associated with increased tumorigenicity via the activation of octamer-binding transcription factor 4 (Oct4)[47,48], a transcription factor that maintains pluripotency[49]. Hypoxia response elements in Oct4 loci have been shown to be bound by HIF-2α, but not by HIF-1α[50]. Notably, Oct4 mRNA has been found to be elevated in patients with class 1 disease[9]. An intriguing speculation, therefore, is that class 2 mutations do not result in sufficient HIF-2α activation to overcome repression of HIF-2α during embryogenesis, while class 1 mutations result in sufficient HIF-2α activation via attenuation of pVHL-mediated degradation to transactivate HIF-2α targets, such as Oct4, necessary for PPGL pathogenesis. Concordantly, a subset of class 1 mutations likely occur during embryogenesis despite being sporadic in nature, due to the presence of multiple neuroendocrine tumors of distinct cellular lineages (PPGL, somatostatinoma) and congenital polycythemia in patients with class 1a and class 1b diseases (Table 1). Moreover, some patients present with HIF-2α mutations in tissues not affected by disease at detectable but non-heterozygous (i.e. mosaic) levels, which provides further evidence that mutations occur before differentiation of a cell lineage, as to allow multiple tissues to be affected[20]. Our hypothesis is consistent with the observation that *VHL* mutations that specifically cause pheochromocytoma appear to disrupt neuronal culling during embryogenesis via dysregulation of atypical protein kinase C in a HIF-independent process[51]. However, one confounding observation is that patients with class 1c disease develop PPGL later in life, usually present with only one neuroendocrine tumor, and do not

develop polycythemia (Supplementary Data 1). We speculate that this phenomenon may be related to the degree of systemic hypoxia a patient experiences. In patients with germline mutations in *SDHD*, which encodes succinate dehydrogenase subunit D, living at higher altitudes was associated with an increased likelihood of developing multiple PPGL tumors[52]. Further investigation is required to determine how HIF-2α activation results in the transformation of chromaffin and paraganglia cells and to delineate the possible role of Oct4 in HIF-2α-driven PPGL tumorigenesis.

Here, we reveal the molecular basis underlying the broad class segregation emerging in HIF-2α-driven disease in which *EPAS1* mutations that cause significant disturbance to the HIF-2α-pVHL interaction interface are associated with class 1 disease while those causing a mild disturbance are associated with class 2 disease. A corollary is that PPGL pathogenesis observed in class 1 disease would require a higher HIF-2α dose than polycythemia, which appears to only require a mild increase in HIF-2α stability as observed with class 2 mutations. Notably, the structure-guided information presented here would be powerful in predicting the broad class phenotype of de novo *EPAS1* mutations in patients.

## Methods

**Prediction of missense mutation effect**. Missense mutations of *EPAS1* (UniProt ascension number: Q99814; Genbank transcript ID: NM_001430; Protein Ensembl ENSP ID: ENSP00000263734) identified in patients with disease were analyzed using the Polyphen-2[53], SIFT[54], MutationTaster2[55], and PROVEAN[56] prediction software. SIFT and PROVEAN predictions were simultaneously determined using PROVEAN Protein Batch (http://provean.jcvi.org; accessed 9 August 2017). A threshold score of -4.1 and 0.05 was used for PROVEAN and SIFT analysis, respectively. A threshold score of −4.1 has been shown to result in increased specificity when using the PROVEAN software[56].

**Alignment of amino acid sequences**. HIF-1α and HIF-2α sequences from mouse (*M. musculus*), flamingo (*P. ruber ruber*), frog (*X. tropicalis*), zebra fish (*D. rerio*), epaulette shark (*H. ocellatum*), fly (*D. melanogaster*), worm (*C. elegans*), and anemone (*E. pallida*) were identified with the protein BLAST algorithm using human HIF-1α (GI:4504385) and HIF-2α (GI:1805268) as the search sequence. Amino acid sequences centered around the primary hydroxylation site were aligned using Clustal Omega[57].

**Plasmids**. The following plasmids have been previously described; pcDNA3-3xFLAG-VHL₃₀(full-length; residues 1–213), pcNDA3-HA-VHL₃₀[58], pcDNA3-HA-HIF2A, pcDNA3-HA-PHD2, pACYCDuet-1 plasmid encoding untagged elongin B and elongin C (17–112), pGEX-4T-1-GST-VHL₁₉ (54–213)[59], and pcDNA3-HA-HIF2A P531A[60]. Mutant pcDNA3-HA-HIF2A full-length constructs were generated via site-directed mutagenesis. pcDNA3-3xFLAG-HIF2A ODD domain (390–554) constructs were sub-cloned via PCR. pET-46 HIS₆-PHD2 plasmid was cloned via ligation independent methods. Primers used for cloning are listed in Supplementary Table 3.

**Antibodies**. Anti-HA antibody was obtained from Cell Signalling Technology (C29F4; 1:2000 dilution western blot). Anti-elongin B (11447; 1:500 dilution ELISA) and Anti-HIS (8036; 1:1000 dilution western blot) were obtained from Santa Cruz. Anti-tubulin (T5168; 1:5000 western blot), anti-vinculin (V9264, 1:10,000 dilution western blot) and anti-FLAG (F1804; 1:5000 western blot; 1:1000 dilution immunoprecipitation) were obtained from Sigma-Aldrich.

**Peptides**. HIF-2α peptides, with N-terminal biotinylation and C-terminal amidation modifications, were custom synthesized by Genscript. For pVHL interaction studies, a WT HIF-2α peptide containing amino acid residues 523–541 was used: ELDLETLA[Hyp]YIPMDGEDFQ (Hyp denotes hydroxylated proline). For hydroxylation assays, or studies involving the L542P mutant, the following WT HIF-2α peptide, containing amino acid residues 523-542, was used: ELDLETLA[Hyp/P]YIPMDGEDFQL. All peptides were reconstituted to 2 mg/mL, as measured by A₂₈₀, using sterile DMSO, aliquoted, and stored at −80 °C.

**Protein expression and purification**. HIS₆-PHD2 (181–426) was expressed in BL21(DE3) *E. coli* cells (Novagen, Cat. No. 69450) and purified according to a published protocol[61]. Cell cultures were grown to OD₆₀₀ = 0.8 and induced with a final concentration of 0.5 mM IPTG for 3.5 h at 37 °C. Bacterial pellets were resuspended in buffer B (40 mM TRIS-HCl pH 7.9, 500 mM NaCl) supplemented

with 5 mM imidazole and lysed using a cell disruption unit at a pressure of 30 kPSI. Cell lysate was cleared via centrifugation (40,700 g for 40 min). Cleared lysate was applied to a column of nickel sepharose resin (Qiagen). The column was rinsed with buffer B supplemented with 5 mM imidazole and washed with buffer B supplemented with 60 mM imidazole. Protein was eluted with buffer B supplemented with 1 M imidazole. Protein solution was concentrated using a 4-mL centrifugal concentrator with a molecular weight cut-off of 10,000 Da (Pall corporation). The concentrated protein solution was purified on a Superdex 200 10/300 size exclusion chromatography column equilibrated with 50 mM Tris-HCl pH 7.5. Purity was confirmed via SDS-PAGE analysis. Aliquots of protein were frozen at −80 °C at a concentration of 2 mg/mL prior to experimentation.

BL21(DE3) *E. coli* cells were co-transformed with a construct encoding N-terminal GST-tagged pVHL₁₉ (54–213) and a dual expression construct encoding untagged elongin B (full-length, 1–118) and elongin C (17–112). 1 L cultures of bacteria were grown to an OD₆₀₀ of approximately 0.6 and expression of the VBC complex was induced with a final concentration of 1 mM IPTG for 3.5 h at a temperature of 37 °C. Bacterial pellets were resuspended in buffer A (20 mM HEPES pH 7.4, 200 mM NaCl) freshly supplemented with 10 mM DTT and lysed using a cell disruption unit at a pressure of 30 kPSI. Cell lysate was cleared via centrifugation (34,000 g for 40 min). Cleared lysate was applied to a column of glutathione sepharose resin (GE Life Sciences). The column was washed with buffer A and protein was eluted with buffer A supplemented with 10 mM reduced glutathione. Protein solution was concentrated using a 4-mL centrifugal concentrator with a molecular weight cut-off of 3500 Da (Pall corporation). The concentrated protein solution was purified on a Superdex 200 10/300 size exclusion chromatography column equilibrated with 20 mM HEPES pH 7.4, 200 mM NaCl, and 1 mM DTT. The monomeric VBC complex was diluted to a concentration of 0.7 mg/mL, which was empirically determined to prevent further aggregation of the protein. The GST-tag was cleaved via incubation with 1 U thrombin per mg of VBC protein at 4 °C. Near complete cleavage of the GST-tag was achieved after incubation for 60 h. The protein solution was applied to a glutathione sepharose column to remove free GST. The protein solution was concentrated as discussed above and size exclusion chromatography was employed to purify soluble, monomeric VBC complex and exchange into a buffer of 5 mM HEPES pH 7.4, 200 mM NaCl, and 1 mM DTT. Purity was confirmed via SDS-PAGE analysis. Aliquots of protein were frozen at −80 °C at a concentration between 1–2 mg/mL prior to biochemical and biophysical experimentation.

**Crystallization**. Freshly purified VBC complex (see above) was incubated with a 2-molar excess of HIF-2αOH peptide for 30 min on ice. Subsequently, the tetrameric complex was concentrated to 10 mg/mL. Sparse matrix screening yielded small crystals in 0.1 M bis-tris pH 6.5, 0.2 M Li₂SO₄, and 25% (w/v) PEG 3350 (condition 23 in the Microlytics MCSG-2 screen). Initially, crystals could not be reproduced in either hanging or sitting drop experiments. However, spherulites obtained in 0.1 M bicine pH 9.0, 20% (w/v) PEG 6000 (condition 30, MCSG-2), when used as seeds allowed for crystal growth in 0.1 M Bis-Tris pH 6.5, 0.2 M Li₂SO₄, 25% (w/v) PEG 3350. Growth of large, singular, diffraction quality crystals was achieved in sitting drops incubated at 20 °C, with a protein concentration of 8 mg/mL, and the use of seed as an additive.

**Data collection and structure determination**. Crystals were cryo-protected with perfluoropolyether oil (Hampton Research) prior to flash cooling to 100 K. X-ray data was subsequently collected on beamline 08ID-1 at the Canadian Light Source using a Pilatus 6 M detector. XDS[62] was used to index and scale crystallographic data. The crystals diffracted to 2.0 Å resolution and belonged to the space group P4₃2₁2 with one complex per asymmetric unit. The structure was determined via molecular replacement using the coordinates for VBC (PDB code: 1lm8)[38]. Coot[63] and Phenix[64] were used to carry out iterative cycles of manual building and refinement, respectively. MOLPROBITY and Coot were used to validate the geometry and model in the structures. Residues 523-526 and 541 of the HIF-2α peptide are disordered and not modeled, while the sidechain of D539 is disordered and built as a Cβ stub. No Ramachandran outliers are present in the final model. Data collection and refinement statistics are presented in Supplementary Table 2. The interface between the HIF-2α peptide and pVHL complex was analyzed via Protein Interfaces, Surfaces and Assemblies (PISA)[65]. PISA was used to calculate the buried area percentage. CLICK was used to calculate RMSD of Cα atoms[66]. Pymol was used to render all structural representations.

**ELISA**. A 96-well streptavidin-coated plate (Quidel) was hydrated with ~200 μL of buffer A per well. Biotinylated HIF-2αOH peptide was diluted to 5 μg/mL in blocking solution (buffer A supplemented with 0.5% (w/v) BSA, 0.02% (v/v) Tween-20, and 1 mM DTT. 0.5 μg of biotinylated HIF2αOH peptide was immobilized in each well. Wells were washed 5x with wash buffer (buffer A supplemented with 0.1% (v/v) Tween-20) followed by a 2 h incubation at room temperatures with 400 μL of blocking solution. 100 μL of VBC complex, at various concentrations (1 μg/mL, 5 μg/mL, and 25 μg/mL), was incubated for 2h at room temperature and then washed 5x with wash buffer. 100 μL of primary anti-elongin B mAb diluted (1:500) in blocking solution was incubated overnight at 4 °C. The plates were washed 5x prior to incubation with 100 μL of secondary goat anti-rabbit IgG mAb diluted 1:1000 in blocking solution for 1 h at room temperature.

Following the final wash step, the plates were developed according to manufacturer's recommendations with ready-to-use TMB1 reagent (Thermo-fisher, Cat. No. 34028) for 20 min before being quenched with $2 N H_2SO_4$. Absorbance at 450 nm was measured using a plate reader. Three independent experiments were performed in duplicate. Absorbance from a blank with no immobilized peptide was subtracted from other values. A two-way ANOVA with Bonferroni post hoc test was conducted to test for statistical significance. A $p$ value below 0.05 was considered significant.

**In vitro binding assay**. The in vitro pVHL-HIF-2α binding assay was performed according to a previously published protocol[67]. HA-VHL$_{30}$ was expressed in a transcription and translation (TNT) rabbit reticulocyte lysate system (Promega, Cat. No. L1170) and incubated with 2 µg of biotinylated HIF-2αOH peptide (WT or mutant), immobilized on streptavidin agarose beads, in either 500 µL of EBC buffer (50 mM Tris-HCl pH 8.0, 120 mM NaCl, 0.5% (v/v) NP-40; stringent buffer conditions) or buffer A supplemented with 0.02% (v/v) Tween-20 (mild buffer conditions) for 2 h at 4 °C. Following incubation, beads were either washed 5x with NETN buffer (20 mM Tris-HCl pH 8.0, 100 mM NaCl, 1 mM EDTA, 0.5% (v/v) NP-40; stringent) or buffer A supplemented with 0.1% (v/v) Tween-20 (mild). Biotinylated peptide was pulled down via streptavidin agarose beads and protein was eluted by boiling beads in sample buffer. Protein levels of HA-VHL$_{30}$ were determined by western blotting. Uncropped western blots are provided in Supplementary Figure 8.

5 µg of biotinylated HIF-2α peptide (WT or mutant) was immobilized on streptavidin agarose beads and incubated with a 50 µg/µL solution of purified HIS$_6$-PHD2 (181–426) in 50 mM Tris-HCl pH 7.5, 0.005% (v/v) Tween-20 buffer supplemented with or without 1 mM MnCl$_2$ and 1 mM NOG, for 2 h at 4 °C. Beads were washed 2x with the same buffer. Biotinylated peptide was pulled down via streptavidin agarose beads and protein was eluted by boiling beads in sample buffer. Protein levels of HIS$_6$-PHD2 were determined by western blotting. Uncropped western blots are provided in Supplementary Figure 8.

**In vitro hydroxylation assay**. The in vitro pVHL-HIF-2α hydroxylation assay was performed according to a previously published protocol[67]. 5 µg of biotinylated HIF-2α peptide was immobilized on streptavidin-agarose beads. Subsequently, peptides were incubated with 15 µg/mL HIS$_6$-PHD2 (181–426) in 500 µL of 40 mM HEPES pH 7.4, 80 mM KCl, 5 mM a-ketoglutarate, 2 mM ascorbic acid, 100 µM FeCl$_2$ tetrahydrate solution at room temperature for 1 h. FeCl$_2$ was stored under anoxic conditions to prevent oxidation. Following hydroxylation, beads were washed 5x with EBC buffer and peptides were incubated with IVTT HA-VHL$_{30}$ under the stringent buffer conditions described above.

**Biolayer interferometry**. The binding affinities of VBC to the hydroxylated HIF2αOH peptide was measured by biolayer interferometry using the BLItz system (Pall ForteBio). A 50 µg/mL solution of biotinylated HIF-2α peptide was prepared in kinetics buffer (buffer A supplemented with 1 mM DTT, 0.02% (v/v) Tween-20, and 0.5% (w/v) BSA) and immobilized onto streptavidin (SA)-coated biolayer interferometry (BLI) biosensors (Pall ForteBio, Cat. No. 18-5019) over 120 s. Multiple concentrations of purified VBC complex were diluted in kinetics buffer and allowed to associate with immobilized peptide over 120 s. Subsequently, the SA-BLI probe was immersed into kinetics buffer for 120 s to allow for dissociation. The data were analyzed, step corrected, reference corrected, and fit to a global 1:1 binding model. $K_d$, $k_a$, and $k_d$ were calculated using the BLItz Pro software.

**Cell culture and transfections**. HEK293a cells were obtained from American Type Culture Collection (ATCC). Cells were maintained in DMEM (Invitrogen) supplemented with 10% (v/v) fetal bovine serum (Wisent) and were grown at 37 °C in a humidified, 5% CO2 atmosphere. The indicated plasmids were transiently transfected into HEK293a cells using polyethylenimine.

**Immunoprecipitation**. 48 h post transfection, cells were harvested in EBC lysis buffer containing protease inhibitors (Roche). Lysates were immunoprecipitated at 4 °C using the indicated antibodies along with protein A-Sepharose (Repligen) for 1.5 h. In degradation sensitive assays, immunoprecipitation was performed in the presence of MG132 proteasomal inhibitor. Following immunoprecipitation, the bound proteins were washed 5x with NETN buffer. The samples were eluted by boiling in sample buffer and protein levels were visualized via western blotting. Uncropped western blots are provided in Supplementary Figure 8.

**Data availability**. The structure and coordinates have been deposited in the Protein Data Bank with accession coded 6BVB. Other data are available from the corresponding author upon reasonable request.

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

## Acknowledgements

We thank the members of Ohh and Lee labs for their critical comments and discussions. We also thank Wenguang He for generating the pET-46 HIS$_6$-PHD2 plasmid and Farshad Cyrus Azimi for assistance with protocol optimization. This work was supported by grants from the Canadian Institutes of Health Research (CIHR) (MOP-133694 to J.E.L.; MOP-77718 and 136978 to M.O.). D.T. is a recipient of CIHR-Vanier Canada Graduate Scholarship and Ontario Graduate Scholarship. This work is based upon X-ray data remotely collected at beamline 08ID-1 at the Canadian Light Source, which is supported by the Canada Foundation for Innovation, Natural Sciences and Engineering Council of Canada, University of Saskatchewan, Government of Saskatchewan, Western Economic Diversification Canada, National Research Council Canada, and CIHR.

## Author contributions

M.O. and D.T. conceptualized the study design. D.T. performed the biochemical, bio-physical, and structural experiments. C.M.R. performed the cell line based experiments. J.E.L. aided in designing and performing the structural experiments. D.T., J.E.L. and M.O. wrote the manuscript.

## Additional information

**Competing interests:** The authors declare no competing interests.

