## [Peer Review File · Nature Communications]

Reviewers' comments:

Reviewer #1 (Remarks to the Author):

In this manuscript, authors solved the structure of complex composed of HIF-2 α peptide and pVHL-Elongin B-Elongin C (VBC) heterotrimeric complex, and tried to find genotype-phenotype correlations in HIF-2 α -driven disease by biochemical analyses. While the findings in this manuscript seem to be potentially intriguing, I have several concerns on this manuscript. Furthermore, as authors state on line 210-213, the disease classes associated with HIF-2 α mutations could not be absolutely predicted solely by determining affinity for pVHL. Therefore, I think this manuscript does not help to find genotype-phenotype correlations.

Major points:

1. The authors examined the affinity between pVHL and HIF-2 α mutants, and tried to find genotype-phenotype correlations. I think it is also important to examine the interaction between HIF-2 α mutants and HIF-1 β to check transcriptional activity of these mutant HIF-2 α /HIF-1 β heterodimer. (By IP-Western and BLI).
2. The authors concluded that some of class 1 mutation affects PHD2-mediated hydroxylation of HIF-2 α (Fig. 4). I think it is better to examine the interaction between HIF-2 α mutants and PHD2 utilizing IP-Western and BLI. Also, negative controls (non-hydroxylated HIF-2 α peptide without PHD2 treatment) should be included in Fig. 4A and 4B.
3. Is there any correlation between class1 or 2 mutation and stability of HIF-2 α ? Cycloheximide chase experiment utilizing 786-O cells (or other pVHL-deficient cells) (with or without pVHL expression) is easy to compare the stabilities of HIF-2 α mutants.
4. The balance of HIF-1 α and HIF-2 α might affect symptoms. As reviewed, the genes regulated by HIF-1 α might be biased toward renal carcinoma "suppressors" and that regulated by HIF-2 α might be biased to renal carcinoma "oncoproteins" (Chuan Shen, William G. Kaelin Jr., The VHL/HIF axis in clear cell renal carcinoma, *Seminars in Cancer Biology* 23 (2013) 18–25). If the interaction between HIF-2 α and pVHL is disrupted by HIF-2 α mutation, pVHL can target HIF-1 α more efficiently?

Minor point:

1. Addition of corresponding HIF-1 α sequence in Fig. 1B would improve the manuscript.
2. The authors just repeat the result section in the discussion section. It could be written clearly and concisely.
3. It is better to change the title because the authors could not reveal genotype-phenotype correlations.
4. It is better to include error bars in Fig. 3D.

Reviewer #2 (Remarks to the Author):

Tarade et al. used structural and biochemical techniques to explain how mutations in HIF-2 α affect binding to VHL and consequently cause disease states. They solved the structure of the VHL-EloBC complex with a HIF-2 α peptide, containing the majority of residues found mutated in patients. The authors then used an in vitro pull-down assay and biolayer interferometry to confirm defects in VHL binding by most HIF-2 α mutants found in patients. Furthermore, they identified mutations (A530V and L532P) which indirectly impair VHL binding by disrupting the hydroxylation of P531.

My major concern is that there is a glaring lack of comparison with the multiple available HIF-1 α /VHL/EloBC structures. HIF-1 α and 2 α have very high sequence identity, especially in the region bound to VHL. Judging from Figure 2, the structures look very similar. It is not clear whether this work brings substantial advancement to what has already been known from the HIF-1 α structures. A detailed structural comparison, with figures and RMSDs of the interface residues, should be carried out to answer the question that the authors raised "it is not known whether it shares the

same binding mode as HIF-1 α , limiting the utility of existing HIF-1 α -pVHL structures". Giving the apparent high level of similarities, the authors should tone down the novelty of the work.

The paper is for the most part is written clearly. There are some modifications to the figures and text that may improve clarity.

1. Fig. 2b is very crowded and difficult to parse. The distance labels may not be necessary.
2. In line 169, Fig. 3b should be referenced instead of 4b.
3. The figure legends for 2b and 2c are switched.
4. In line 205, R108 should be R107.
5. The figure legend title for Fig. S6 is written twice.

Reviewer #3 (Remarks to the Author):

In the manuscript titled "Crystal structure of HIF-2 α peptide bound to pVHL-EloB-EloC complex reveals a structural basis for genotype-phenotype correlations in HIF-2 α -driven disease" prepared by Tarade et al., the authors investigated the structure of a HIF-2 α mutant peptide in the VBC complex, trying to explain the relationship between HIF2A mutations and clinical disease manifestations/outcomes. The manuscript provides interesting insights to understanding the functional changes in mutant HIF-2 α that could explain various clinical phenotypes (here almost solely between classes 1 and 2). The authors used several sophisticated and well thought out techniques and approaches.

The authors need to think carefully about whether their results can fully explain genotype-phenotype correlations outlined in this study. Whereas their results could aptly apply to predicting differences between classes 1 and 2 (and here phenotypic outcomes), this is less convincing for various clinical phenotypes inside class 1, where data is somewhat lacking. This reviewer's comment is also well supported by the authors' abstract/summary which focuses on the differences between classes 1 and 2 but not within class 1. Therefore, the title of the manuscript may need to be altered. Nevertheless, their conclusion/suggestion that one could predict genotype-phenotype correlations based on the mutation type is clinically very important and very useful. Although, to a certain degree it is limited, since only some patients present with mosaicism and most HIF2A mutations causing a tumor development are somatic.

Several concerns are listed below:

Arguable characterization of mutation subgroups: The author investigated 66 reported HIF2A mutation cases, stratifying them into subclasses based on their clinical manifestations (very well done and thought out). However, the proposed classification may have some limitations as correctly mentioned in the manuscript:

- i. One genotype could lead to very diverse clinical manifestations. For example, the A530V variant was first identified in complete syndrome, and later in one case with neuroendocrine tumor but no polycythemia. Similarly, the variant D539N was detected in a patient carrying full syndrome with ocular manifestation, and another patient with PPGL, ocular manifestation, but no somatostatinoma (authors fairly mentioned it in the manuscript).
- ii. Various clinical manifestations of this disease/syndrome are related to somatic mosaicism that takes place at a very early stage of embryonic development. The disease spectrum may be largely determined by the timing of mutation and the cellular lineages that are hit by the variant (the authors tackled this information in the manuscript). Please discuss any further thoughts about it.

Further comments:

How does PREVEAN score (also described in the Figure 1D) help the understanding of this disease, predicted pathogenicity among various mutations.... See your text on page 3, In 115-120. This information, especially the presentation of the Figure 1D, is unclear. Thus, the page 4, In 112-115, authors announced 61.1% of class 1 mutants are deleterious and only 33.3% of class 2 mutants as deleterious. However, from the Figure 1D, the least negative PREVEAN scored mutants (less

deleterious mutants) in class 1 are much more than class 2. So, the reviewer is not convinced that there is "a trend towards increased predicted pathogenicity among class 1 mutants".

Page 4: In 158: "Class 1 HIF-2 mutations have more deleterious effect on pVHL binding"...this sentence is "incomplete". Using "more" means that there needs to be a comparison with "something else" here "than class 2..."

What do authors mean by 'pVHL affinity'? Affinity to what? pVHL binds to various structures. This needs to be well described in the text (see page 4, also pVHL binding vs pVHL affinity is often used interchangeably).

What is the view of authors on HIF-2 α ubiquitination and degradation as validation of VBC recognition? The author applied in vitro binding assays to investigate the affinity between HIF-2 α peptides with VHL, followed by co-immunoprecipitation as further validation. The consequent protein ubiquitination, degradation, and HRE transactivation could be further evaluated.

It would be useful to acknowledge any shortcomings of the present study, notably, predicting phenotype-genotype in the class 1.

In the Figure 1A, it is extremely difficult to distinguish the different classes on the horizontal axis. In the Figure 3B, there is a dramatic decreasing affinity between the HIF-2 -L529P and VHL. Why are there no binding affinity changes of HIF-2 -L529P and VBC?

Minor:

PPGLs are catecholamine producing, not secreting, tumors.

"Bilateral presentation" is incorrect clinical terminology here; avoid it and re-write appropriately.

Oxygen is only one of the co-substrates for PHDs, re-write.

Oxford and US English are mixed together.

References: Titles of journal are in small letters; needs reformatting.

RESPONSES TO REVIEWERS

Reviewer No. 1 (Remarks to the Author):

In this manuscript, authors solved the structure of complex composed of HIF-2 α peptide and pVHL-Elongin B-Elongin C (VBC) heterotrimeric complex, and tried to find genotype-phenotype correlations in HIF-2 α -driven disease by biochemical analyses. While the findings in this manuscript seem to be potentially intriguing, I have several concerns on this manuscript. Furthermore, as authors state on line 210-213 (now **244-247**), the disease classes associated with HIF-2 α mutations could not be absolutely predicted solely by determining affinity for pVHL. Therefore, I think this manuscript does not help to find genotype-phenotype correlations.

RESPONSE: We thank the reviewer for the critical assessment of our manuscript. With respect to the above general comment, while we appreciate the reviewer underscoring that HIF-2 α disease classes cannot be *absolutely* predicted solely by determining binding affinity to pVHL, we would like to reiterate the reasons behind this statement and the lines of investigation that followed to interrogate the limitations of strictly using synthetically hydroxylated disease-associated HIF-2 α peptides for predicting broad genotype-phenotype correlations. In addition, the full statement was, ‘although BLI kinetic analysis revealed a distinguishable kinetic trend between class 1 and class 2 mutations, the disease class associated with HIF-2 α mutations could not be absolutely predicted solely by determining affinity for pVHL due to outliers like A530V’. Briefly, we demonstrate that A530V impedes PHD2-mediated hydroxylation of P531, which consequently attenuates HIF-2 α binding affinity for pVHL in a range consistent with class 1 mutants. Furthermore, based on the structural information gained from the crystal structure of HIF-2 α peptide bound to pVHL-EloB-EloC complex, we show that class 1 mutations cluster on residues contacting pVHL while class 2 mutations generally localize to a non-contacting ‘kink’ region. Biophysical experiments showed that class 1 mutations disrupt pVHL affinity more adversely than class 2 mutations directly or indirectly via impeding PHD2-mediated hydroxylation as in A530V mutant. Therefore, we show that combination of these biophysical assays can be used to differentiate between HIF-2 α mutations that cause class 1 versus class 2 HIF-2 α -driven disease.

Major points:

1. The authors examined the affinity between pVHL and HIF-2 α mutants, and tried to find genotype-phenotype correlations. I think it is also important to examine the interaction between HIF-2 α mutants and HIF-1 β to check transcriptional activity of these mutant HIF-2 α /HIF-1 β heterodimer. (By IP-Western and BLI).

RESPONSE: We respectfully disagree with this suggestion for the following reasons. The HIF-2 α domains responsible for DNA binding (basic helix loop helix) and dimerization with HIF-1 β (Pas-A and Pas-B) span amino acid residues 3-361 and function independently of the oxygen-dependent degradation (ODD) domain (Wu et al., 2015). To date, purification of recombinant full-length HIF- α has not been attainable, making a technique like BLI ill-suited for measuring the effect of ODD domain mutations, which cluster around amino acid residue 531. Furthermore, ChIP studies with WT and L529P HIF-2 α revealed that both bind to HRE elements with identical affinity, strongly suggesting that mutations within the distant ODD domain do not impact the PAS or bHLH domains (Yang et al., 2013). In other words, this line of investigation would not be informative in imparting genotype-phenotype correlations.

2. The authors concluded that some of class 1 mutation affects PHD2-mediated hydroxylation of HIF-2 α (Fig. 4). I think it is better to examine the interaction between HIF-2 α mutants and PHD2 utilizing IP-Western and BLI. Also, negative controls (non-hydroxylated HIF-2 α peptide without PHD2 treatment) should be included in Fig. 4A and 4B.

RESPONSE: Done. We performed binding assays with recombinant PHD2 to test the reviewer's suggestion that disease class may also be correlated with degree of PHD2 binding. However, a major technical hurdle with performing binding assay with enzymes is overcoming the inherently transient nature of the interaction. Concordantly, our earlier experiments showed that even the HIF-2 α WT peptides failed to pull-down recombinant PHD2 protein. In an effort to stabilize the interaction between HIF-2 α peptide and PHD2, we incorporated a technique that was used during co-crystallization of HIF-1 α peptide with PHD2 (Chowdhury et al., 2009). As PHD2 normally forms a stable complex with co-factors Fe²⁺ and α -ketoglutarate (α KG), we formed a complex of PHD2 with Mn²⁺ and *N*-oxalylglycine (NOG), which mimic iron and α KG, respectively, but do not allow for enzymatic turnover. During BLI experiments, the addition of Mn²⁺ and NOG to the buffer decreased the K_d by roughly 1000-fold (from mM to μ M). However, determination of accurate affinity constants with BLI was not possible due to moderate aggregation of PHD2 on the biosensors at concentrations necessary for obtaining adequate signal (**New Fig. X for the Reviewer**). This resulted in a poor fit between actual binding behaviour and modelled binding behaviour. The trend, however, of increased binding between HIF-2 α peptide and PHD2 when incubated with Mn²⁺ and NOG suggested that a pull-down experiment may be a viable alternative. Ultimately, we observed binding between HIF-2 α WT and PHD2 under steady-state conditions (**New Fig. S6a**). All tested mutations resulted in a large decrease in affinity of HIF-2 α peptide for PHD2, regardless of associated disease class (**New Fig. S6a**). These results were further validated in HEK293a cells via co-IP with ectopically expressed HIF-2 α ODD harbouring various disease-associated mutations and PHD2 (**New Fig. S6b**). These results taken together suggest that affinity for PHD2 under steady-state conditions cannot accurately differentiate disease class associated with a particular HIF-2 α mutation.

Thus, the hydroxylation assay used in **Fig. 5** (previously Fig. 4) is a more direct and reliable measure of hydroxylation than steady-state or kinetic binding experiments with PHD2. Having extensively validated that the A530V HIF-2 α mutant peptide can bind to pVHL (either recombinant VBC complex or *in vitro* transcribed and translated VHL₃₀) when synthetically hydroxylated, the diminished interaction between A530V peptide and pVHL when prior hydroxylation via PHD2 is required clearly shows that the defect lies with PHD2-mediated hydroxylation. This outcome is also clearly seen with the L542P mutant, which cannot bind PHD2. Synthetically hydroxylated L542P peptide can bind to pVHL. However, if a non-hydroxylated peptide is instead utilized, incubation with recombinant PHD2 cannot rescue interaction with pVHL, suggesting that this mutation inhibits PHD2-mediated hydroxylation. This type of hydroxylation assay has been previously described as one of only two direct measures of PHD activity (Hewitson et al., 2007).

Negative controls (non-hydroxylated HIF-2 α peptide without PHD2, non-hydroxylated HIF-2 α peptide without VHL) have been included in **Fig. 5a and 5b** (previously Fig. 4a and 4b). The overall trend remained unchanged and adjustments to our methodology for the further optimization of the hydroxylation assay have been outlined in the revised manuscript.

3. Is there any correlation between class1 or 2 mutation and stability of HIF-2 α ? Cycloheximide chase experiment utilizing 786-O cells (or other pVHL-deficient cells) (with or without pVHL

expression) is easy to compare the stabilities of HIF-2 α mutants.

RESPONSE: We have performed these assays, but found them to be uninformative. For example, *in vitro* and cellular ubiquitylation assays, which represent a process preceding proteasome-mediated degradation and thus more direct than a protein half-life analysis, showed most efficient ubiquitylation of HIF-2 α ODD WT than disease-causing HIF-2 α ODD mutants, but failed to reliably distinguish between class 1 and 2 HIF-2 α mutants (**New Fig. Y for the Reviewer**). In other words, while these suggested assays would be informative in validating the general gain-of-function effect of the identified HIF-2 α mutants, they would not be sufficiently sensitive to differentiate between class 1 and class 2 diseases.

4. The balance of HIF-1 α and HIF-2 α might affect symptoms. As reviewed, the genes regulated by HIF-1 α might be biased toward renal carcinoma “suppressors” and that regulated by HIF-2 α might be biased to renal carcinoma “oncoproteins” (Chuan Shen, William G. Kaelin Jr., The VHL/HIF axis in clear cell renal carcinoma, *Seminars in Cancer Biology* 23 (2013) 18–25). If the interaction between HIF-2 α and pVHL is disrupted by HIF-2 α mutation, pVHL can target HIF-1 α more efficiently?

RESPONSE: The reviewer raises an intriguing notion. It should, however, be noted that although it has been previously proposed that HIF-1 α functions as a RCC tumour suppressor (evidence being that HIF-1 α is inactivated in a number of RCC cell lines), more recent data obtained from transgenic mouse models suggest that both HIF-1 α and HIF-2 α contribute to tumourigenesis. In particular, it is proposed that HIF-1 α is crucial for metabolic re-programming and tumour initiation (Fu et al., 2011; Schonenberger et al., 2016). Moreover, it is currently unknown whether HIF-1 α acts as a tumour suppressor in chromaffin or paraganglia cells, relevant for PPGL and the current study. Nevertheless, we agree that the notion of increased efficiency in HIF-1 α degradation in the presence of reduced HIF-2 α and pVHL interaction is possible. However, we respectfully feel that determining experimentally whether or how the interplay between HIF-1 α and the various HIF-2 α mutations contributes to the genotype-phenotype correlations in HIF-2 α driven disease is beyond the scope of this manuscript.

Minor point:

1. Addition of corresponding HIF-1 α sequence in Fig. 1B would improve the manuscript.

RESPONSE: Done. An alignment of HIF-1 α and HIF-2 α protein from various species has been included (**New Fig. 2a**).

2. The authors just repeat the result section in the discussion section. It could be written clearly and concisely.

RESPONSE: Done. We have substantially reworked the discussion section to minimize repetition.

3. It is better to change the title because the authors could not reveal genotype-phenotype correlations.

RESPONSE: Done. Please see our response to the reviewer's opening comment. Based on the structural information gained from HIF-2 α peptide-VBC co-crystal, we showed that HIF-2 α mutations that cause class 1 disease can be differentiated from those that cause class 2 disease using biophysical assays. However, the study does not provide an evidence-based explanation of the various subclasses within class 1 disease including those class 1 disease-causing HIF-2 α mutations that are associated with more than one subclass (see **Fig. 1c**). Thus, we have modified the title of our manuscript from '...structural basis for genotype-phenotype correlations...' to '...structural basis for *broad* genotype-phenotype correlations...' to reflect the limitation of our study.

4. It is better to include error bars in Fig. 3D.

RESPONSE: Done (now **Fig. 4d**). Please note that the standard errors associated with K_d , k_d , and k_{off} are listed in the accompanying **Table 2**. Furthermore, we have also included the individual data points in **Fig. 4a** along with standard deviation.

Reviewer No. 2 (Remarks to the Author):

Tarade et al. used structural and biochemical techniques to explain how mutations in HIF-2 α affect binding to VHL and consequently cause disease states. They solved the structure of the VHL-EloBC complex with a HIF-2 α peptide, containing the majority of residues found mutated in patients. The authors then used an in vitro pull-down assay and biolayer interferometry to confirm defects in VHL binding by most HIF-2 α mutants found in patients. Furthermore, they identified mutations (A530V and L542P) which indirectly impair VHL binding by disrupting the hydroxylation of P531.

My major concern is that there is a glaring lack of comparison with the multiple available HIF-1 α /VHL/EloBC structures. HIF-1 α and 2 α have very high sequence identity, especially in the region bound to VHL. Judging from Figure 2, the structures look very similar. It is not clear whether this work brings substantial advancement to what has already been known from the HIF-1 α structures. A detailed structural comparison, with figures and RMSDs of the interface residues, should be carried out to answer the question that the authors raised “it is not known whether it shares the same binding mode as HIF-1 α , limiting the utility of existing HIF-1 α -pVHL structures”. Giving the apparent high level of similarities, the authors should tone down the novelty of the work.

RESPONSE: We thank the reviewer for raising this important point and the reasonable suggestion of tempering the novelty of our study. We have now conducted a comparison between HIF-1 α and HIF-2 α binding motifs (**New Fig. 2**). Notably, we highlight the sequence similarities between HIF-1 α and HIF-2 α while discussing two evolutionarily conserved residues that are different. These differences (HIF-1 α M561/HIF-2 α T528 and the insertion of HIF-2 α G537; **New Fig. 2a and c**) prompted us to pursue an unbiased crystallographic approach to co-crystallize HIF-2 α peptide with VHL as we did not formally know whether HIF-2 α would bind pVHL with a binding motif similar to HIF-1 α . Most salient was the observation that HIF-2 α G537, not conserved in HIF-1 α , is the most commonly mutated residue in HIF-2 α -driven disease (**New Fig. 2a and d**). Thus, the structural significance of HIF-2 α G537 or corresponding mutational substitutions in HIF-2 α -driven disease could not be learned from the existing HIF-1 α -pVHL structures. Furthermore, although HIF-1 α and HIF-2 α peptides bind to pVHL with a similar binding motif, the RMSD between HIF-2 α and HIF-1 α peptides bound to pVHL range from 0.78 (1LM8) to 0.85 Å (1LQB). The main difference is the increased sinusoidal nature of the ‘kink’ region in HIF-2 α in comparison to HIF-1 α binding motif (**New Fig. 2d**). The manuscript has been revised accordingly to better explain our rationale and the findings with respect to HIF-1 α and HIF-2 α binding motifs while tempering the novelty (see revised text, **p4, lines 131-165**).

The paper is for the most part written clearly. There are some modifications to the figures and text that may improve clarity.

1. Fig. 2b is very crowded and difficult to parse. The distance labels may not be necessary.

RESPONSE: Agreed. The H-bond distances have been removed.

2. In line 169, Fig. 3b should be referenced instead of 4b.

RESPONSE: Corrected. We apologize for this oversight.

3. The figure legends for 2b and 2c are switched.

RESPONSE: Corrected. We apologize for this oversight.

4. In line 205, R108 should be R107.

RESPONSE: In our model, pVHL R108 is modelled as a CB-stub (the side chain is disordered). We predict that mutation of HIF2 α G537 to Arg or Trp would result in steric clash with R108, not R107. The figure has been redesigned for better clarity (i.e., R107 is no longer labeled while R108 backbone is labeled).

5. The figure legend title for Fig. S6 is written twice.

RESPONSE: Corrected.

Reviewer No. 3 (Remarks to the Author):

In the manuscript titled "Crystal structure of HIF-2 α peptide bound to pVHL-EloB-EloC complex reveals a structural basis for genotype-phenotype correlations in HIF-2 α -driven disease" prepared by Tarade et al., the authors investigated the structure of a HIF-2 α mutant peptide in the VBC complex, trying to explain the relationship between HIF2A mutations and clinical disease manifestations/outcomes. The manuscript provides interesting insights to understanding the functional changes in mutant HIF-2 α that could explain various clinical phenotypes (here almost solely between classes 1 and 2). The authors used several sophisticated and well thought out techniques and approaches.

The authors need to think carefully about whether their results can fully explain genotype-phenotype correlations outlined in this study. Whereas their results could aptly apply to predicting differences between classes 1 and 2 (and here phenotypic outcomes), this is less convincing for various clinical phenotypes inside class 1, where data is somewhat lacking. This reviewer's comment is also well supported by the authors' abstract/summary which focuses on the differences between classes 1 and 2 but not within class 1. Therefore, the title of the manuscript may need to be altered. Nevertheless, their conclusion/suggestion that one could predict genotype-phenotype correlations based on the mutation type is clinically very important and very useful. Although, to a certain degree it is limited, since only some patients present with mosaicism and most HIF2A mutations causing a tumor development are somatic. Several concerns are listed below:

Arguable characterization of mutation subgroups: The author investigated 66 reported HIF2A mutation cases, stratifying them into subclasses based on their clinical manifestations (very well done and thought out). However, the proposed classification may have some limitations as correctly mentioned in the manuscript:

1. One genotype could lead to very diverse clinical manifestations. For example, the A530V variant was first identified in complete syndrome, and later in one case with neuroendocrine tumor but no polycythemia. Similarly, the variant D539N was detected in a patient carrying full syndrome with ocular manifestation, and another patient with PPGL, ocular manifestation, but no somatostatinoma (authors fairly mentioned it in the manuscript).

RESPONSE: Firstly, we thank the reviewer for the overall positive assessment of our work. Secondly, regarding the above astute comment, we absolutely agree that our present study does not provide a structural or mechanistic explanation underlying subclasses within class 1, including those class 1 disease-causing HIF-2 α mutations that are associated with more than one subclass (see **Fig. 1c**). We have included an expanded discussion on the limitations of our study, underscoring that the structural information provided in our study, while providing biophysical explanation between class 1 and class 2 diseases, is insufficient for explaining or predicting subclasses within class 1. We have also included a discussion on our thoughts related to the development of multiple subclasses of class 1 disease via single HIF-2 α mutation, inferring the significance of mutation timing and cellular lineages (see **p9, lines 360-394** for detail). Moreover, we have modified the title of our manuscript from '...structural basis for genotype-phenotype correlations...' to '...structural basis for *broad* genotype-phenotype correlations...' to reflect the limitation of our study.

2. Various clinical manifestations of this disease/syndrome are related to somatic mosaicism that takes place at a very early stage of embryonic development. The disease spectrum may be largely determined by the timing of mutation and the cellular lineages that are hit by the variant (the authors tackled this information in the manuscript). Please discuss any further thoughts about it.

RESPONSE: Done. We are in agreement with the reviewer on the various clinical manifestations related to somatic mosaicism. We have further expanded our thoughts on the possible role of mutation timing and cellular lineages on the emergence of class 1 disease subclasses (see **p9, lines 360-394** for detail).

Further comments:

3. How does PROVEAN score (also described in the Figure 1D) help the understanding of this disease, predicted pathogenicity among various mutations.... See your text on page 3, In 115-120. This information, especially the presentation of the Figure 1D, is unclear. Thus, the page 4, In 112-115, authors announced 61.1% of class 1 mutants are deleterious and only 33.3% of class 2 mutants as deleterious. However, from the Figure 1D, the least negative PROVEAN scored mutants (less deleterious mutants) in class 1 are much more than class 2. So, the reviewer is not convinced that there is "a trend towards increased predicted pathogenicity among class 1 mutants".

RESPONSE: We thank the reviewer for raising this point. Our argument regarding the trend toward pathogenicity seen among class 1 mutants is based on the aforementioned percentage of HIF-2 α mutations meeting the threshold for pathogenicity in each disease class, with nearly twice the percentage of class 1 mutations meeting that threshold. That being said, as we discussed in the text, PROVEAN proved unable to absolutely segregate class 1 and class 2 mutations in large part due to a cluster of class 1 mutants that are predicted to be 'benign'. As the reviewer correctly pointed out, there is a greater percentage of benign HIF-2 α mutations in class 1 disease than in class 2 disease. The text has been revised to better reflect this observation (see **p3, lines 121-128**). Overall, our main argument regarding the use of *in silico* mutation characterisation tools is that they are inadequate at segregating HIF-2 α mutations of different disease classes.

4. Page 4: In 158: "Class 1 HIF-2 mutations have more deleterious effect on pVHL binding"...this sentence is "incomplete". Using "more" means that there needs to be a comparison with "something else" here "than class 2..."

RESPONSE: Agreed. This has been corrected (**p5, line 189**).

5. What do authors mean by 'pVHL affinity'? Affinity to what? pVHL binds to various structures. This needs to be well described in the text (see page 4, also pVHL binding vs pVHL affinity is often used interchangeably).

RESPONSE: We apologize for the ambiguity. We have better clarified in the text. We are also aware of pVHL having multiple binding partners and substrates. In this study, we specifically mean the strength of interaction between HIF-2 α peptide or protein (depending on the experiment) and pVHL when using the term 'pVHL affinity' (see **p5, lines 192-193**). We have clarified any text describing pVHL affinity to HIF-2 α . We use 'pVHL binding' when discussing the physical interaction between HIF-2 α and pVHL without regard for the strength of interaction or in cases where there is a lack of binding kinetic information.

6. What is the view of authors on HIF-2 α ubiquitination and degradation as validation of VBC recognition? The author applied *in vitro* binding assays to investigate the affinity between HIF-2 α peptides with VHL, followed by co-immunoprecipitation as further validation. The consequent protein ubiquitination, degradation, and HRE transactivation could be further evaluated.

RESPONSE: We offer two thoughts. First, this work is based on the structural information gained from the HIF-2 α peptide-VBC crystal that suggests potential biophysical explanation as to why certain mutations are associated with class 1 disease while others are associated with class 2 HIF-2 α disease. The most direct consequence of HIF-2 α mutations was hypothesized to influence the binding kinetics between HIF-2 α and pVHL, and thus, we examined this interaction using various approaches designed to measure the strength of pVHL-HIF-2 α interaction. BLI approach was most informative as it was quantitative and providing both association and dissociation constants. Second, while the consequent ubiquitylation, degradation, and HRE transactivation could be further evaluated, these events are subsequent to the most direct aforementioned binding measurements. Perhaps more importantly, the available assays designed to measure the level of ubiquitylation, degradation (i.e., protein half-life analysis) and HRE transactivation (depends on ectopic HIF-2 α expression) would not be able differentiate between class 1 and 2 mutations.

We also know this empirically because we have performed these assays and as anticipated, these approaches were not informative for the purpose of genotype-phenotype study. The most revealing of these assays were the *in vitro* and cell-based HIF-2 α ODD ubiquitylation assay (**New Fig. Y for the Reviewer**). Although HIF-2 α ODD WT was most efficiently ubiquitylated, the distinction between the ubiquitylation profiles of class 1 versus class 2 HIF-2 α ODD mutants was very difficult to discern. We do believe that these experiments (ubiquitylation, degradation and HRE transactivation) would be useful for initial validation of novel disease-associated HIF-2 α mutations as causing a gain-of-function. However, more quantitative experiments such as BLI would be necessary for discriminating between class 1 and class 2 mutations.

7. It would be useful to acknowledge any shortcomings of the present study, notably, predicting phenotype-genotype in the class 1.

RESPONSE: Done. Please see our response to this reviewer's above comment no. 1.

8. In the Figure 1A, it is extremely difficult to distinguish the different classes on the horizontal axis.

RESPONSE: Agreed. We now provide an inset of amino acids 515-550 where the majority of HIF-2 α mutations have been identified (revised **Fig. 1a**).

9. In the Figure 3B, there is a dramatic decreasing affinity between the HIF-2 -L529P and VHL. Why are there no binding affinity changes of HIF-2 -L529P and VBC?

RESPONSE: The dramatic decrease in affinity of HIF-2 α L529P for pVHL is observed in our BLI experiments where the K_d of the L529P mutant (558 nM) for VBC is markedly higher than WT HIF-2 α (196 nM) for VBC. Thus, we argue that the L529P and A530T class 1 mutations have a similar effect on pVHL affinity when binding is analysed via BLI and pull-down. However, we concede that the L529P mutation does not have a statistically significant effect when the VBC-HIF2 α interaction was examined via ELISA. However, in light of the revealed K_d values obtained via BLI experimentation, the ELISA approach was clearly not sensitive enough to detect the changes in pVHL affinity to the studied HIF-2 α mutations (see **p8, lines 335-344**).

Minor:

1. PPGLs are catecholamine producing, not secreting, tumors.

RESPONSE: Corrected.

2. "Bilateral presentation" is incorrect clinical terminology here; avoid it and re-write appropriately.

RESPONSE: Corrected.

3. Oxygen is only one of the co-substrates for PHDs, re-write.

RESPONSE: Agreed. Corrected. A full list of PHD co-substrates is now noted in the text.

4. Oxford and US English are mixed together.

RESPONSE: Corrected for Oxford English.

5. References: Titles of journal are in small letters; needs reformatting.

RESPONSE: Done.

Figure X. Biolayer interferometry studies of HIF-2 α peptide binding to PHD2. Biolayer interferometry kinetic analysis of HIS6-PHD2(181-426) binding to biotinylated HIF-2 α peptides (523-542). Biotinylated peptides were coupled to streptavidin-coated biosensors and monitored for binding to PHD2 at the indicated concentrations in the presence or absence of 1 mM MnCl₂ and 1 mM *N*-oxalylglycine. The data were analyzed based on a 1:1 binding model using the BLitz Pro software with the fitted curves shown as gray lines. Sensorgrams are representative of two experiments conducted with independently purified proteins.

Figure Y. Ubiquitylation assays do not differentiate class 1 and class HIF-2α mutants. (A) 3xFLAG-HIF-2α oxygen-dependent degradation (ODD) domains were expressed with a rabbit reticulocyte transcription and translation system. ODD protein was incubated with or without exogenous ubiquitin (1 mg/mL) in the presence of a reaction mix (2 mM ATP, 40 mM creatine phosphate, 0.5 μg/μL creatine phosphate, 25 μg/mL ubiquitin aldehyde, 100 μg/mL HIS6-PHD2(181-426) *in vitro* transcribed and translated HA-VHL₃₀, 5 mM MgCl₂, 2 mM KCl, 20 mM Tris-HCL pH 7.4) for two hours at 30 °C. ODD protein was immunoprecipitated (IP) with an anti-FLAG antibody subsequent to visualization via immunoblotting (IB). (B) Hek293A cells transfected with 3xFLAG-HIF-2α ODD constructs with or without Myc-ubiquitin (MycUb). ODD protein was immunoprecipitated ed with anti-FLAG and levels of Myc-ubiquitin were visualized via immunoblotting. FLAG and vinculin levels in whole cell extracts were determined via immunoblotting.

REVIEWERS' COMMENTS:

Reviewer #1 (Remarks to the Author):

Comment:

The authors proposed that the affinity between mutant HIF-2 α and pVHL is the determinant for the neuroendocrine tumors (class 1 disease) and polycythemia (class 2 disease). However, they do not show the stabilities of mutant HIF-2 α and the stabilization of HIF-2 α by the mutation.

Furthermore, the transcriptional activities of mutant HIF-2 α have never been examined. Therefore, it is not clear whether proposed class 1 and class 2 mutations are the determinant of each disease. The revised manuscript is insufficient to show the relationships between HIF-2 α mutations and different types of tumors, and I cannot recommend publication in Nature communications. I added individual comments below.

Reviewer No. 1 (Remarks to the Author):

In this manuscript, authors solved the structure of complex composed of HIF-2 α peptide and pVHL-Elongin B-Elongin C (VBC) heterotrimeric complex, and tried to find genotype-phenotype correlations in HIF-2 α -driven disease by biochemical analyses. While the findings in this manuscript seem to be potentially intriguing, I have several concerns on this manuscript. Furthermore, as authors state on line 210-213 (now 244-247), the disease classes associated with HIF-2 α mutations could not be absolutely predicted solely by determining affinity for pVHL. Therefore, I think this manuscript does not help to find genotype-phenotype correlations.

RESPONSE: We thank the reviewer for the critical assessment of our manuscript. With respect to the above general comment, while we appreciate the reviewer underscoring that HIF-2 α disease classes cannot be absolutely predicted solely by determining binding affinity to pVHL, we would like to reiterate the reasons behind this statement and the lines of investigation that followed to interrogate the limitations of strictly using synthetically hydroxylated disease-associated HIF-2 α peptides for predicting broad genotype-phenotype correlations. In addition, the full statement was, 'although BLI kinetic analysis revealed a distinguishable kinetic trend between class 1 and class 2 mutations, the disease class associated with HIF-2 α mutations could not be absolutely predicted solely by determining affinity for pVHL due to outliers like A530V'. Briefly, we demonstrate that A530V impedes PHD2-mediated hydroxylation of P531, which consequently attenuates HIF-2 α binding affinity for pVHL in a range consistent with class 1 mutants. Furthermore, based on the structural information gained from the crystal structure of HIF-2 α peptide bound to pVHL-EloB-EloC complex, we show that class 1 mutations cluster on residues contacting pVHL while class 2 mutations generally localize to a non-contacting 'kink' region. Biophysical experiments showed that class 1 mutations disrupt pVHL affinity more adversely than class 2 mutations directly or indirectly via impeding PHD2-mediated hydroxylation as in A530V mutant. Therefore, we show that combination of these biophysical assays can be used to differentiate between HIF-2 α mutations that cause class 1 versus class 2 HIF-2 α -driven disease.

Major points:

1. The authors examined the affinity between pVHL and HIF-2 α mutants, and tried to find genotype-phenotype correlations. I think it is also important to examine the interaction between HIF-2 α mutants and HIF-1 β to check transcriptional activity of these mutant HIF-2 α /HIF-1 β heterodimer. (By IP-Western and BLI).

RESPONSE: We respectfully disagree with this suggestion for the following reasons. The HIF-2 α domains responsible for DNA binding (basic helix loop helix) and dimerization with HIF-1 β (Pas-A and Pas-B) span amino acid residues 3-361 and function independently of the oxygen-dependent degradation (ODD) domain (Wu et al., 2015). To date, purification of recombinant full-length HIF- α has not been attainable, making a technique like BLI ill-suited for measuring the effect of ODD domain mutations, which cluster around amino acid residue 531. Furthermore, CHIP studies with WT and L529P HIF-2 α revealed that both bind to HRE elements with identical affinity, strongly suggesting that mutations within the distant ODD domain do not impact the PAS or bHLH domains

(Yang et al., 2013). In other words, this line of investigation would not be informative in imparting genotype-phenotype correlations.

Comment:

Authors do not perform IP-Western blotting or other experiment to examine HIF-2 α transcriptional activity, therefore it is not clear whether proposed class 1 and class 2 mutations affect HIF α / β transcriptional activity. As authors explain, if these mutations do not affect transcriptional activity, how these mutations contribute to class1 or class 2 phenotypes is unclear. Therefore, revised manuscript remains insufficient.

2. The authors concluded that some of class 1 mutation affects PHD2-mediated hydroxylation of HIF-2 α (Fig. 4). I think it is better to examine the interaction between HIF-2 α mutants and PHD2 utilizing IP-Western and BLI. Also, negative controls (non-hydroxylated HIF-2 α peptide without PHD2 treatment) should be included in Fig. 4A and 4B.

RESPONSE: Done. We performed binding assays with recombinant PHD2 to test the reviewer's suggestion that disease class may also be correlated with degree of PHD2 binding. However, a major technical hurdle with performing binding assay with enzymes is overcoming the inherently transient nature of the interaction. Concordantly, our earlier experiments showed that even the HIF-2 α WT peptides failed to pull-down recombinant PHD2 protein. In an effort to stabilize the interaction between HIF-2 α peptide and PHD2, we incorporated a technique that was used during co-crystallization of HIF-1 α peptide with PHD2 (Chowdhury et al., 2009). As PHD2 normally forms a stable complex with co-factors Fe²⁺ and α -ketoglutarate (α KG), we formed a complex of PHD2 with Mn²⁺ and N-oxalylglycine (NOG), which mimic iron and α KG, respectively, but do not allow for enzymatic turnover. During BLI experiments, the addition of Mn²⁺ and NOG to the buffer decreased the K_d by roughly 1000-fold (from mM to μ M). However, determination of accurate affinity constants with BLI was not possible due to moderate aggregation of PHD2 on the biosensors at concentrations necessary for obtaining adequate signal (New Fig. X for the Reviewer). This resulted in a poor fit between actual binding behaviour and modelled binding behaviour. The trend, however, of increased binding between HIF-2 α peptide and PHD2 when incubated with Mn²⁺ and NOG suggested that a pull-down experiment may be a viable alternative. Ultimately, we observed binding between HIF-2 α WT and PHD2 under steady-state conditions (New Fig. S6a). All tested mutations resulted in a large decrease in affinity of HIF-2 α peptide for PHD2, regardless of associated disease class (New Fig. S6a). These results were further validated in HEK293a cells via co-IP with ectopically expressed HIF-2 α ODD harbouring various disease-associated mutations and PHD2 (New Fig. S6b). These results taken together suggest that affinity for PHD2 under steady-state conditions cannot accurately differentiate disease class associated with a particular HIF-2 α mutation. Thus, the hydroxylation assay used in Fig. 5 (previously Fig. 4) is a more direct and reliable measure of hydroxylation than steady-state or kinetic binding experiments with PHD2. Having extensively validated that the A530V HIF-2 α mutant peptide can bind to pVHL (either recombinant VBC complex or in vitro transcribed and translated VHL30) when synthetically hydroxylated, the diminished interaction between A530V peptide and pVHL when prior hydroxylation via PHD2 is required clearly shows that the defect lies with PHD2-mediated hydroxylation. This outcome is also clearly seen with the L542P mutant, which cannot bind PHD2. Synthetically hydroxylated L542P peptide can bind to pVHL. However, if a nonhydroxylated peptide is instead utilized, incubation with recombinant PHD2 cannot rescue interaction with pVHL, suggesting that this mutation inhibits PHD2-mediated hydroxylation. This type of hydroxylation assay has been previously described as one of only two direct measures of PHD activity (Hewitson et al., 2007).

Negative controls (non-hydroxylated HIF-2 α peptide without PHD2, non-hydroxylated HIF-2 α peptide without VHL) have been included in Fig. 5a and 5b (previously Fig. 4a and 4b). The overall trend remained unchanged and adjustments to our methodology for the further optimization of the hydroxylation assay have been outlined in the revised manuscript.

3. Is there any correlation between class1 or 2 mutation and stability of HIF-2 α ? Cycloheximide chase experiment utilizing 786-O cells (or other pVHL-deficient cells) (with or without pVHL expression) is easy to compare the stabilities of HIF-2 α mutants.

RESPONSE: We have performed these assays, but found them to be uninformative. For example, in vitro and cellular ubiquitylation assays, which represent a process preceding proteasome-mediated degradation and thus more direct than a protein half-life analysis, showed most efficient ubiquitylation of HIF-2 α ODD WT than disease-causing HIF-2 α ODD mutants, but failed to reliably distinguish between class 1 and 2 HIF-2 α mutants (New Fig. Y for the Reviewer). In other words, while these suggested assays would be informative in validating the general gain-of-function effect of the identified HIF-2 α mutants, they would not be sufficiently sensitive to differentiate between class 1 and class 2 diseases.

Comment:

Since the authors do not show the stabilities of HIF-2 α mutants, it is not clear how these HIF-2 α mutants are involved in the class 1 or class 2 phenotypes. The ubiquitination data shown in the revised manuscript are not sufficient because of the lack of negative control (without pVHL, figure Y(A)) to show the pVHL-dependent ubiquitination. Figure Y(B) is also insufficient. Authors immunoprecipitated with anti-FLAG antibody (HIF-2 α ODD) and immunoblotted with anti-myc antibody (myc-Ub), indicating that we are detecting the ubiquitination of HIF-2 α ODD and HIF-2 α ODD-interacting proteins. This experiment also lacks negative control (without pVHL overexpression).

Another point is that, since different types of ubiquitin chain exist, ubiquitination level does not always correlate with stability of the substrate (in this case HIF-2 α). Therefore, the data comparing the stabilities of HIF-2 α mutants is necessary.

4. The balance of HIF-1 α and HIF-2 α might affect symptoms. As reviewed, the genes regulated by HIF-1 α might be biased toward renal carcinoma "suppressors" and that regulated by HIF-2 α might be biased to renal carcinoma "oncoproteins" (Chuan Shen, William G. Kaelin Jr., The VHL/HIF axis in clear cell renal carcinoma, *Seminars in Cancer Biology* 23 (2013) 18–25). If the interaction between HIF-2 α and pVHL is disrupted by HIF-2 α mutation, pVHL can target HIF-1 α more efficiently?

RESPONSE: The reviewer raises an intriguing notion. It should, however, be noted that although it has been previously proposed that HIF-1 α functions as a RCC tumour suppressor (evidence being that HIF-1 α is inactivated in a number of RCC cell lines), more recent data obtained from transgenic mouse models suggest that both HIF-1 α and HIF-2 α contribute to tumourigenesis. In particular, it is proposed that HIF-1 α is crucial for metabolic re-programming and tumour initiation (Fu et al., 2011; Schonenberger et al., 2016). Moreover, it is currently unknown whether HIF-1 α acts as a tumour suppressor in chromaffin or paraganglia cells, relevant for PPGL and the current study. Nevertheless, we agree that the notion of increased efficiency in HIF-1 α degradation in the presence of reduced HIF-2 α and pVHL interaction is possible. However, we respectfully feel that determining experimentally whether or how the interplay between HIF-1 α and the various HIF-2 α mutations contributes to the genotype phenotype correlations in HIF-2 α driven disease is beyond the scope of this manuscript.

Reviewer #3 (Remarks to the Author):

The authors revised the manuscript according to the Reviewer's comments.

The Reviewer particularly likes addressing shortcomings of the present study, the title change as well as the revised figures and the discussion including the hypothesis related to Oct4. It is also well appreciated that the authors included a recent study related to cardiac abnormalities, PGL and HIF2A mutations.

The Reviewer has only minor comments:

Ln 47: "PPGL tumours" should be "PPGLs"

Ln 51: "sporadic PPGL" should be "sporadic PPGLs"

Ln 64: "paraganglioma" should be "PGL"

Ln 321: "PPGL carcinogenesis" should be "PPGL tumorigenesis"

Figure Y: "Hek293" should be "HEK293"

Reviewer #1 (Remarks to the Author):

The authors proposed that the affinity between mutant HIF-2 α and pVHL is the determinant for the neuroendocrine tumors (class 1 disease) and polycythemia (class 2 disease). However, they do not show the stabilities of mutant HIF-2 α and the stabilization of HIF-2 α by the mutation. Furthermore, the transcriptional activities of mutant HIF-2 α have never been examined. Therefore, it is not clear whether proposed class 1 and class 2 mutations are the determinant of each disease. The revised manuscript is insufficient to show the relationships between HIF-2 α mutations and different types of tumors, and I cannot recommend publication in Nature communications. I added individual comments below.

Comment:

Authors do not perform IP-Western blotting or other experiment to examine HIF-2 α transcriptional activity, therefore it is not clear whether proposed class 1 and class 2 mutations affect HIF α / β transcriptional activity. As authors explain, if these mutations do not affect transcriptional activity, how these mutations contribute to class1 or class 2 phenotypes is unclear. Therefore, revised manuscript remains insufficient.

Response: Perhaps we did not make our original response sufficiently clear. First, we are not suggesting that these mutations would not affect transcriptional output of HIF-2. Rather, the observed differential binding between class 1 and class 2 mutants to pVHL E3 ligase complex would ultimately influence the stability of HIF-2 α and the transcriptional activity of HIF-2. See also our response to comment below. We are simply arguing that these disease-causing mutations on HIF-2 α would not affect heterodimerization with HIF-1 β because they are located well outside of domains responsible for dimerization with HIF-1 β (i.e., Pas-A and Pas-B) or DNA binding (i.e., basic helix loop helix). Thus, any assessment of HIF-2 α disease-associated mutants to binding HIF-1 β , as originally suggested by the reviewer, would be uninformative in imparting genotype-phenotype correlation.

Comment:

Since the authors do not show the stabilities of HIF-2 α mutants, it is not clear how these HIF-2 α mutants are involved in the class 1 or class 2 phenotypes. The ubiquitination data shown in the revised manuscript are not sufficient because of the lack of negative control (without pVHL, figure Y(A)) to show the pVHL-dependent ubiquitination. Figure Y(B) is also insufficient. Authors immunoprecipitated with anti-FLAG antibody (HIF-2 α ODD) and immunoblotted with anti-myc antibody (myc-Ub), indicating that we are detecting the ubiquitination of HIF-2 α ODD and HIF-2 α ODD-interacting proteins. This experiment also lacks negative control (without pVHL overexpression). Another point is that, since different types of ubiquitin chain exist, ubiquitination level does not always correlate with stability of the substrate (in this case HIF-2 α). Therefore, the data comparing the stabilities of HIF-2 α mutants is necessary.

Response: We show here via solving the crystal structure of HIF-2 α peptide bound to pVHL-elongin B-elongin C complex that class 1 and class 2 HIF-2 α mutations segregate topographically and are predicted to have differential binding affinity to pVHL. We make the argument that these differences are significant in that the broad disease phenotype (i.e. class 1 versus class 2) can be predicted based on the binding kinetics. While it is reasonable and logical to suggest that these differences in binding kinetics would translate to corresponding HIF-2 α stability and ultimately HIF-2 transcriptional output, we make the argument that these downstream events (i.e., post binding to pVHL complex events) do not improve the predictive power of quantitative binding kinetic measurements that correlate with broad class 1 and 2 disease phenotypes. Again, we are not suggesting that these HIF-2 α mutations would not affect the stability or transcriptional activity of HIF-2 α .

In regards to the specifics, in Fig. Ya, the rabbit reticulocyte contains endogenous pVHL. pVHL is supplemented to boost signal. Thus, a negative control lacking pVHL is not possible in this experimental

system. Instead, we provide a P405A/P531A HIF2 α as a negative control, which is a HIF2 α that cannot be ubiquitinated by pVHL complex. Thus, any bands present in the P405A/P531A lane represent either un-ubiquitinated HIF-2 α , IgG bands, or HIF-2 α that is ubiquitinated by other E3 complexes. In Fig. Yb, we do not over-express pVHL. Rather, all ubiquitination is attributed to endogenous pVHL. It is well-taken that by immunoprecipitating anti-FLAG (HIF-2 α ODD) may result in precipitation of HIF-2 α interacting partners that are also ubiquitinated. However, the strong anti-Myc (ubiquitin) signal obtained with WT HIF-2 α ODD but not mutant HIF-2 α ODD constructs strongly suggests that HIF-2 α mutation results in decreased ubiquitination, which is in agreement with our extensive experimentation showing that HIF-2 α mutations results in decreased hydroxylation and/or pVHL binding. We also agree that ubiquitination is not always associated with proteasomal-mediated degradation, although the canonical HIF- α pathway does indeed culminate in such a process.

--

Reviewer #3 (Remarks to the Author):

The authors revised the manuscript according to the Reviewer's comments.

The Reviewer particularly likes addressing shortcomings of the present study, the title change as well as the revised figures and the discussion including the hypothesis related to Oct4. It is also well appreciated that the authors included a recent study related to cardiac abnormalities, PGL and HIF2A mutations.

The Reviewer has only minor comments:

Ln 47: "PPGL tumours" should be "PPGLs"

Response: Done.

Ln 51: "sporadic PPGL" should be "sporadic PPGLs"

Response: Done.

Ln 64: "paraganglioma" should be "PGL"

Response: Done.

Ln 321: "PPGL carcinogenesis" should be "PPGL tumorigenesis"

Response: Done.

Figure Y: "Hek293" should be "HEK293"

Response: Done.